# Spatiotemporal single-cell RNA sequencing of developing chicken hearts identifies interplay between cellular differentiation and morphogenesis

Madhav Mantri [1,2,4], Gaetano J. Scuderi[1,4], Roozbeh Abedini-Nassab[1,3], Michael F. Z. Wang[1,2], David McKellar[1], Hao Shi [1], Benjamin Grodner[1], Jonathan T. Butcher [1✉] & Iwijn De Vlaminck [1✉]

Single-cell RNA sequencing is a powerful tool to study developmental biology but does not preserve spatial information about tissue morphology and cellular interactions. Here, we combine single-cell and spatial transcriptomics with algorithms for data integration to study the development of the chicken heart from the early to late four-chambered heart stage. We create a census of the diverse cellular lineages in developing hearts, their spatial organization, and their interactions during development. Spatial mapping of differentiation transitions in cardiac lineages defines transcriptional differences between epithelial and mesenchymal cells within the epicardial lineage. Using spatially resolved expression analysis, we identify anatomically restricted expression programs, including expression of genes implicated in congenital heart disease. Last, we discover a persistent enrichment of the small, secreted peptide, thymosin beta-4, throughout coronary vascular development. Overall, our study identifies an intricate interplay between cellular differentiation and morphogenesis.

[1] Nancy E. and Peter C. Meinig School of Biomedical Engineering, Cornell University, Ithaca, NY, USA. [2] Computational Biology Ph.D. Program, Cornell University, Ithaca, NY, USA. [3] Department of Engineering, University of Neyshabur, Neyshabur, Iran. [4] These authors contributed equally: Madhav Mantri, Gaetano J. Scuderi. ✉email: jtb47@cornell.edu; vlaminck@cornell.edu

The heart is the first fully functional organ to develop and is vital for embryogenesis[1]. Cardiogenesis involves heterogeneous cell populations from multiple lineages that spatiotemporally interact to drive cardiac fate decisions[2]. The heterogeneity of cell types during cardiac development makes it difficult to study cardiac fate decisions using traditional developmental biology techniques. Single-cell RNA-sequencing (scRNA-seq) has been used to study the cellular mechanisms involved in heart development, but does not preserve spatial information, and does not enable studies of the complex interplay between cellular differentiation and morphogenesis. Here, we combined spatially resolved RNA-seq with high-throughput scRNA-seq to study the spatiotemporal interactions and regulatory programs that drive embryonic chicken heart development. Current spatial transcriptomics approaches lack single-cell resolution, which we addressed here using approaches to integrate high-throughput spatial and single-cell transcriptomic data.

Chicken embryos were used as a cardiogenesis model system since the ex-utero development in an egg allows unique access to early cardiogenesis stages and the chicken heart anatomy resembles many aspects of human heart anatomy[3]. We generated over 22,000 single-cell transcriptomes across four key Hamburger-Hamilton ventricular development stages (HH21-HH24, HH30-HH31, HH35-HH36, and HH40). The data encompasses common and rare cell types, including progenitor and mature cell types from multiple lineages. In addition, we performed spatially resolved RNA-seq on a total of 12 heart tissue sections at the same four stages.

In this study, the combination of single-cell and spatial transcriptomics uniquely enables us to unravel cellular interactions that drive cardiogenesis and reconstruct a high-resolution, spatially resolved gene expression census of developmental cardiac lineages. We characterize and spatially resolve progenitor and differentiated cell types, identify stage-specific transcriptional programs and cellular interactions, reconstruct differentiation lineages, and delineate important regulatory programs in cardiac development. We integrate scRNA-seq and spatial RNA-seq data using an anchor-based method to predict cell type annotations for spatially resolved transcriptomes. We use these cell-type predictions to construct proximity maps identifying changes in local cellular environments and uncovered spatially restricted regulatory programs. We furthermore construct a similarity map between single-cell and spatial transcriptomes, which enables us to spatially map lineage-associated differentiation trajectories within the tissue. This analysis identifies transcriptional differences between epithelial and mesenchymal cells, further clarifies the differentiation transitions within the epicardial lineage, and points to the utility of spatiotemporal single-cell RNA sequencing to study cardiogenesis.

## Results

**Spatially resolved single-cell transcriptomics census of developing embryonic chicken hearts**. To study the complex interplay between differentiation and morphogenesis during cardiac development, we combined single-cell and spatial transcriptomics. We profiled four key Hamburger-Hamilton ventricular development stages of the chicken heart: (i) day 4 (HH21-HH24, whole ventricles), corresponding to the early chamber formation stage during the initiation of ventricular septation and only trabeculated myocardium, (ii) day 7 (HH30-HH31, left and right ventricles), one of the earliest stages of cardiac four-chamber formation with ventricular septation almost complete but the myocardium containing mostly fenestrated trabeculated sheets, (iii) day 10 (HH35-HH36, left and right ventricles), a mid-four chamber development stage with septation complete and mostly compact myocardium, and (iv) day 14 (~HH40, left and right ventricles), a late four-

chamber development stage with completely compact myocardium and the main events of cardiogenesis essentially complete[3].

To perform scRNA-seq (10x Genomics Chromium platform), we enzymatically digested cardiac ventricular tissue into single-cell suspensions (Fig. 1a, see "Methods" section). The pooling of cells from up to 72 embryonic hearts enabled scRNA-seq on ventricular tissue at the early stages of development. To perform spatial transcriptomics (10x Genomics Visium platform), we cryosectioned 10 μm coronal tissue slices (chamber view) from embryonic hearts at the same four stages (Fig. 1a, See "Methods" section). We generated single-cell transcriptomic data for 22,315 cells and spatial transcriptomics data for 12 tissue sections covering over 6800 barcoded spots (Supplementary Fig. 1a and Supplementary Fig. 1b). We found that spatial transcriptomes collected at the same developmental stage were strongly correlated (Pearson correlation; $R > 0.98$, Supplementary Fig. 1d), and that spatial transcriptomes and single-cell transcriptomes collected at the same developmental stage were strongly correlated (Pearson correlation; $R$ 0.88-0.91, Supplementary Fig. 1e). The combination of scRNA-seq and spatial transcriptomics uniquely enabled us to spatially resolve cell-type-specific gene expression in cardiac tissues.

To analyze single-cell transcriptomes, we filtered and pre-processed the data (see "Methods" section), performed batch correction using scanorama[4] (Supplementary Fig. 1c), performed dimensionality reduction and cell clustering, and then visualized the data by Uniform Manifold Approximation and Projection (UMAP, see "Methods" section). This analysis identified 15 distinct cardiac cell type clusters (Fig. 1b). We used canonical marker gene and differential gene expression analysis to assign cell types to cell clusters (Supplementary Data 1, Fig. 1) and identified diverse cell clusters from myocardial, endocardial, and epicardial cardiac lineages in the ventricles (Fig. 1d, Supplementary Fig. 1f). In addition to cardiac cell types, we detected a small number of erythrocytes, macrophages, and dendritic cells. Last, we identified a unique heterogeneous population of cells that express high levels of thymosin beta-4 (TMSB4X). A detailed overview of the cell-types identified is provided in the supplement (Supplementary Data 1).

Standalone analysis of the spatial transcriptomic data identified anatomical regions with differential transcriptional programs. To spatially resolve cell populations, the spatial transcriptomics data was integrated with the scRNA-seq data using Seurat-v3 anchor-based integration[5,6]. This approach first identifies anchors between datasets, which represent pairwise correspondences between elements in the two datasets that appear to originate from the same biological state. The anchors are then used to harmonize the datasets by learning a joint structure with canonical correlation analysis and to transfer annotation information from one dataset to the other. Every spot in the spatial data could be considered a weighted mix of cell-types identified by scRNA-seq. We used the prediction scores from label transfer to obtain weights for each of the scRNA-seq-derived cell types for each spot (Fig. 1c, Supplementary Fig. 2a–o, see "Methods" section). To understand the spatial organization of cell types in broad anatomical regions, spots were labeled as cell types with a maximum prediction score and visualized on H&E stained images of respective stages (Fig. 1c).

Cell-type prediction scores for spatial transcriptomes were further used to estimate the abundance of pairs of specific cell types (see "Methods" section). As proximity is a necessity for physical interactions between two or more cells, these cell-type proximity maps can be used to guide the discovery of interactions between cell types from the same or different lineages. We constructed proximity maps for all cardiac cell type pairs and

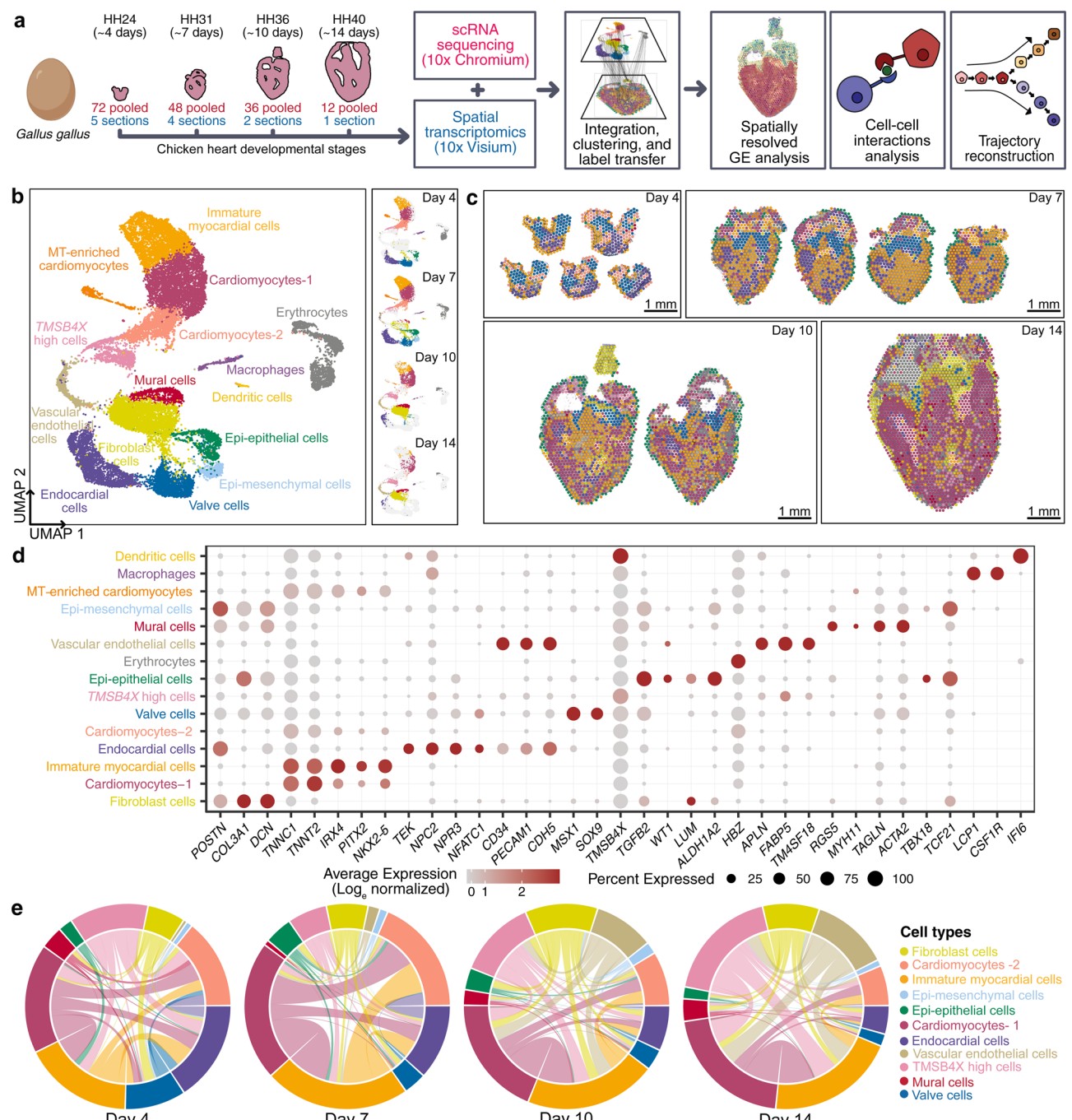

**Fig. 1 Spatially resolved single-cell transcriptomic atlas of developing embryonic chicken hearts. a** Experimental workflow and analysis for single-cell RNA-seq and spatial RNA-seq of embryonic chicken (*Gallus gallus*) hearts at four stages of development. **b** UMAP projection of 22,315 single-cell transcriptomes clustered by gene expression and colored by cell type (Left). UMAP projection of single-cell transcriptomes colored by cell type and split by developmental stage from day 4 to day 14 (Right). **c** Spatial RNA-seq barcoded spots labeled by scRNA-seq cell type with maximum prediction score for four developmental stages. Only spots under tissue sections are shown. **d** Gene expression of cell type-specific markers. The size of the dot represents the percentage of cells in the clusters expressing the marker and the color intensity represents the average expression of the marker in that cluster. **e** Chord diagrams representing cell-type proximity maps showing the degree of colocalization of cell-type pairs within spots in spatial RNA-seq data across developmental stages.

visualized them as chord diagrams (Fig. 1e). We found a significant colocalization of myocardial cells with endocardial cells at day 7 and with vascular endothelial and fibroblast cells at day 10 and day 14. This was expected given that endocardial cells line the trabeculated myocardium at day 7 and that vascular endothelial cells and fibroblasts are present in the compact myocardium by day 10.

**Spatially resolved cardiac lineage analysis**. Single-cell RNA sequencing enables us to profile transcriptional regulation of highly heterogeneous cell populations and facilitates the discovery of genes that identify cell subtypes, or that mark intermediate states during a biological process, as well as bifurcate between two alternative cellular fates. We hypothesized that characterizing differentiation transitions in a spatial context would identify

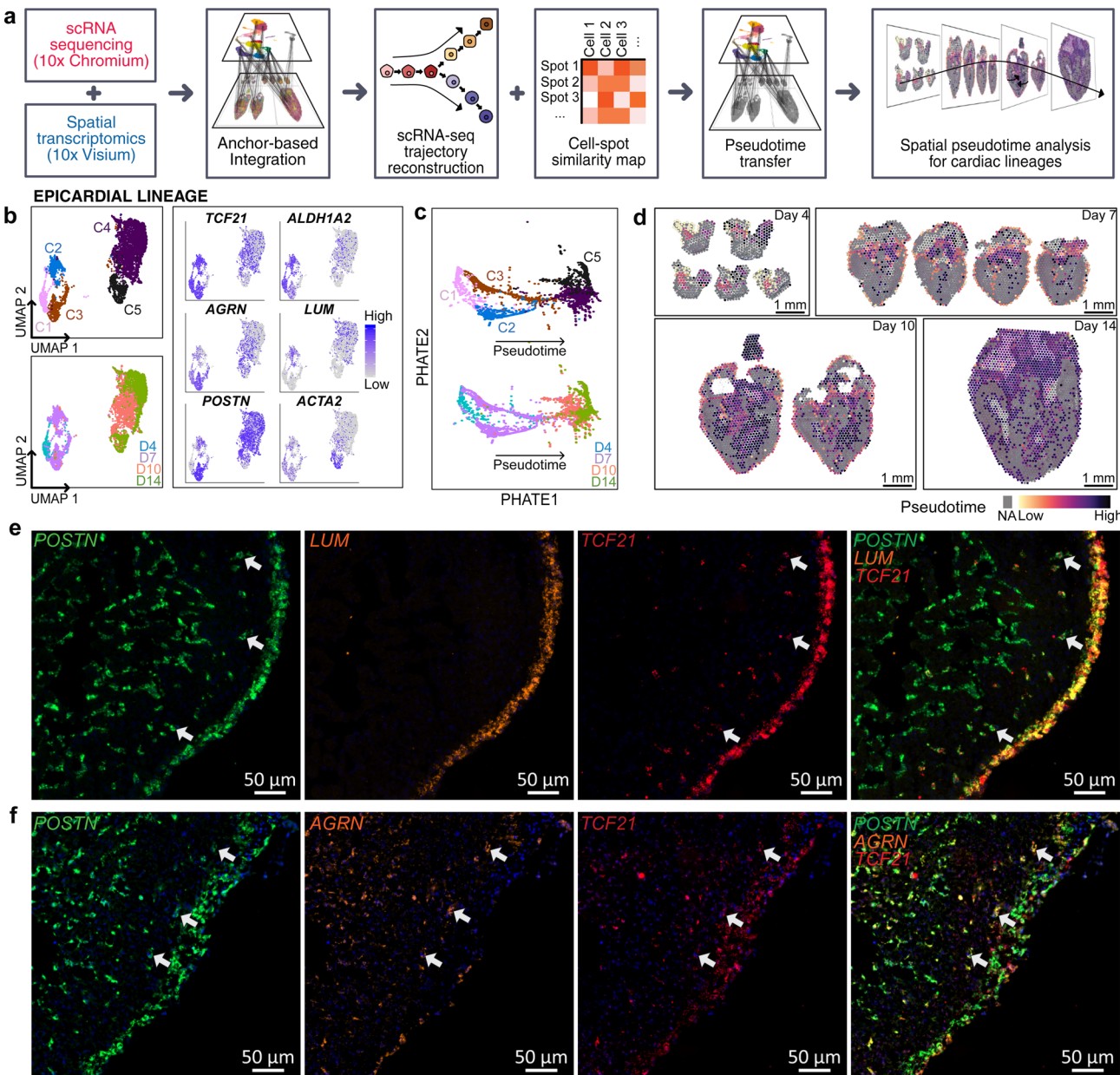

**Fig. 2 Spatiotemporal lineage analysis identifies heterogeneity in cardiac progenitor cells. a** Overview of analysis pipeline for trajectory reconstruction for scRNA-seq and spatial RNA-seq data. **b** UMAP projection of single-cell transcriptomes from individual epicardial lineage cells clustered by gene expression and colored by cell type (left-top) and developmental stage (left-bottom). Feature plots showing expression of gene markers for individual epicardial lineage cells (right). **c** Epicardial lineages visualized by PHATE and labeled by cell type (top) and development stage (bottom). **d** Spatially resolved spatial RNA-seq spot pseudotime for epicardial lineage across developmental stages. Spot pseudotime was estimated using a similarity map between scRNA-seq cells and spatial RNA-seq spots. **e** single-molecule FISH of Day 7 (HH31) ventricular free walls stained for *POSTN* (green), *TCF21* (red), and DAPI (blue) with *LUM* (orange). **f** Single-molecule fluorescent in situ hybridization of Day 7 (HH31) ventricular free walls stained for *POSTN* (green), *TCF21* (red), and DAPI (blue) with *AGRN* (orange). The Blue channel is DAPI. Representative images of three to four biological replicates.

cellular morphogenic events that occur during the differentiation process of ventricular development. To do so, we gathered and reclustered single-cell transcriptomes from the epicardial, endocardial, and myocardial lineages (Supplementary Fig. 1f, Supplementary Data 2), and then reconstructed lineage differentiation trajectories. We used the trajectory reconstruction tools PHATE[7] (Potential of Heat-diffusion for Affinity-based Transition Embedding) and monocle[8] to learn the structure in scRNA-seq data and order cells by developmental state (pseudotime), placing them along a trajectory corresponding to cellular differentiation (Supplementary Note 1). We then spatially projected information

about cellular transitions from scRNA-seq onto the spatial maps by estimating the local PHATE1 dimension, a proxy for developmental time (see "Methods" section, Fig. 2a).

Lineage analysis for epicardial cells identified a rich spatiotemporal differentiation process. The embryonic heart stages analyzed in this study represent key timepoints of epicardial development ranging from before the epicardial-to-mesenchymal transition (EMT) (HH24–day 4), during EMT and epicardial differentiation (HH31–day 7), and during/after epicardial differentiation (HH36–day 10 and HH40–day 14)[3]. We selected epi-epithelial (*TCF21+ WT1+ TEK−*), epi-mesenchymal (*TCF21+ TEK−*),

fibroblast cells (TCF21+ POSTN+ TEK−), and mural cells for epicardial lineage analysis. Clustering of 5,621 epicardial cells from the ventricular free wall identified five main cell clusters: an early epicardial progenitor cells cluster (C1), two intermediate precursor cell clusters (C2 and C3), a fibroblast-like cell cluster (C4), and a mural cell cluster (C5) (Fig. 2b, left and Supplementary Fig. 3a). Cells in cluster C1 were mainly derived from the HH24 (day 4) ventricles and expressed canonical epicardial progenitor markers such as transcription factor 21 (TCF21)[9], t-box transcription factor 18[10] (TBX18), and Wilm's tumor 1 (WT1)[11] (Supplementary Fig. 3b and Supplementary Fig. 3c). They also expressed retinoic acid signaling-related transcript aldehyde dehydrogenase[12,13] (ALDH1A2), a well-known signaling pathway involved in epicardial development (Fig. 2b, left and right). Last, these cells showed an enriched expression of midkine (MDK), another retinoic acid signaling related transcript and growth factor[14], that has not been implicated before in epicardial development (Supplementary Fig. 3b).

At the HH31 (day 7) stage when EMT is underway[3], two main cell clusters were identified (C2 and C3). C2 cells were TCF21+, WT1+, TBX18+, and had an upregulated expression of MDK (Supplementary Fig 3b and Supplementary Fig. 3d, Fig. 2b). In comparison to the C1 cluster, this C2 cluster had an upregulated expression of bone morphogenetic protein 4 (BMP4), which is associated with an epicardial progenitor-like phenotype[15], and lumican (LUM), which is known to be expressed in the outermost epicardial layer of embryonic hearts[16] (Fig. 2b, right, Supplementary Fig. 3e and Supplementary Fig. 3f). In contrast to the C2 cluster, the TCF21+ C3 cell cluster was mostly TBX18− and WT1− (Supplementary Fig 3b, Fig. 2b). Interestingly, these C3 cells expressed high levels of extracellular matrix transcripts implicated in cell migration, including fibronectin (FN1), periostin (POSTN), epidermal growth factor-like domain multiple 7 (EGFL7)[17,18], and agrin (AGRN)[19–22] (Supplementary Fig. 3b, Supplementary Fig. 3g–h and Fig. 2b). This cluster also had upregulated expression of the well-known EMT regulator snail-1 (SNAI1)[23] and was TEK− which is a TIE2 ortholog and an endocardial-derived TCF21+ fibroblast marker[24] (Supplementary Fig. 3b, Fig. 1d). Transferring cell type labels to spatial data suggested that only the epi-epithelial (epicardial subclusters C1 and C2) were spatially restricted to the outermost epicardial layer of the ventricular walls in day 7 and day 10 heart stages (Supplementary Fig. 2a). However, epi-mesenchymal (epicardial subcluster C3) were not spatially restricted to the outer epicardial layer in day 7 hearts, indicating they reside within the myocardium as opposed to the outer epicardial layer (Supplementary Fig. 2b). Therefore, this data further suggests these C3 HH31 day 7 staged cells represent a mesenchymal phenotype of epicardial progenitor-derived cells (EPDCs) that are most likely undergoing EMT and migrating into the myocardium while the C2 HH31 day 7 staged cells are epi-epithelial cells lining the epicardium.

By the day 10 (HH36) heart stage, TCF21+ TEK− epicardial-derived fibroblasts (C4) express collagen type III (COL3A1) and decorin (DCN) and mural cells (C5) expressing ACTA2, MYH11, TAGLN, and RGS5 are present within the myocardium (Fig. 1c, Fig. 2b, Supplementary Fig. 3b). However, some cells in the outermost epicardial layer at the day 10 (HH36) stage still maintain an intermediate undifferentiated phenotype as suggested by our cell type label transferred spatial RNA-seq showing epi-epithelial cells localized to day 10 epicardium (Supplementary Fig. 2a). By the day 14 (HH40) stage, most epicardial layer cells have differentiated to a fibroblast-like phenotype as suggested by their localization to day 14 epicardium, while other fibroblasts and mural cells localize throughout the myocardium as expected (Supplementary Fig. 2c and Supplementary Fig. 2d). PHATE-

based trajectory reconstruction confirmed branching cell fates at day 7 and branch-merging and differentiation at day 10 (Fig. 2c). Additional lineage trajectory analysis by monocle-v2[8] is presented in the supplement (see "Methods" section, Supplementary Fig. 3i–j). Pseudotime correlated well with stages in spatial data and identified significant within-stage variability at days 7 and 10 with a presence of undifferentiated cells in the epicardium lining and differentiated cells in the myocardium (Fig. 2d). Gene ontology enrichment analysis for genes that significantly correlated with pseudotime confirms an increase in extracellular matrix directing cues during epicardial development (Supplementary Fig. 3k).

To support the epicardial lineage results, we performed multiplexed single-molecule fluorescent in situ hybridization (smFISH) assays on day 7 (HH31) heart sections (see "Methods" section). In agreement with our scRNAseq data, we identify TCF21+, POSTN+ cells residing within the compact myocardium of the ventricular free walls that are LUM− suggesting the presence of EPDCs that have undergone EMT and reside within the myocardium (Supplementary Fig. 4a, Fig. 2e). In contrast, cells that line the outermost epicardial layer are TCF21+, POSTN+, and LUM+, (Supplementary Fig. 4a, Fig. 2e). Last, we found that EPDCs upregulate the expression of a variety of key extracellular matrix genes implicated in migration. We performed smFISH to demonstrate that these EPDCs are AGRN+ (Supplementary Fig. 4b, Fig. 2f). Overall, these smFISH results validate the findings from scRNA-seq and spatial RNA-seq and are consistent with the possibility that these cell populations are derived from the epicardial lineage.

For the ventricular endocardial lineage, we did not capture intermediate cell types transitioning to endocardial-derived fibroblasts, endocardial-derived mural cells, or ventricular endocardium-derived vascular endothelial cells, possibly because this transition occurs rapidly in between the stages analyzed in this study[25,26]. We identified two cell clusters within the endocardial lineage: early endocardial cells (C1) from days 4 and 7, and mature endocardial cells (C2) from days 10 and 14 (Supplementary Fig. 5a, left and Supplementary Fig. 5d). The two ventricular endocardial cell subclusters expressed the endocardial marker (NPC2), the differentiated endothelial markers CDH5, PECAM1, PODXL[27], ENG[28], and the retinoic acid signaling related transcript RARRES1[29] (Supplementary Fig. 5a, right). Endocardial cells lined the ventricular chambers of all heart stages in the spatial RNA-seq data, helping to confirm their ventricular endocardial origin (Supplementary Fig. 2e). PHATE based trajectory and pseudotime analysis further confirmed these endocardial cell phenotypes across stages (Supplementary Fig. 5b–e).

In the myocardial lineage, we identified three clusters: an immature myocardial cell cluster (C1) predominantly containing cardiomyocytes from day 4 and 7, a mature cardiomyocyte cell cluster (C2) from mostly day 10 and 14, and a transcriptionally less active cardiomyocyte—a cluster (C3) from day 4 and 7 (Supplementary Fig. 5f and Supplementary Fig. 5j, Fig. 1d). All cells expressed mature cardiomyocyte markers like cardiac troponin (TNNC1), while immature myocardial cells differentially expressed the progenitor markers NKX2-5[30], PITX2[31], and IRX4[32]. The differentiated cardiomyocytes were enriched in myosin light chain-10 (MYL10) and myoglobin (MB) (Supplementary Fig. 5f, right, and Supplementary Fig. 5i). PHATE-based trajectory reconstruction of the myocardial lineage confirmed a progression from an immature phenotype (C1) to a more mature phenotype (C2), but the transcriptionally less active cardiomyocyte-like cells (C3) did not follow this trajectory (Supplementary Fig. 5g and Supplementary Fig. 5h). These trajectory analysis results corroborate the differentiation

trajectory of cardiomyocytes in human fetal hearts reported recently[33].

**Spatiotemporally resolved local cellular heterogeneity in developing cardiac tissue.** To examine spatial transcriptional differences within the embryonic cardiac ventricles, we performed unsupervised clustering of spatial RNA-seq spots and labeled the clusters by anatomical region based on their location in the tissue. Using this analysis, we identified distinct spatial clusters derived from ventricles, atria, valves, and the outflow tract but also distinct layers of ventricular regions including epicardium, compact and trabecular myocardium regions, and endocardium (Fig. 3a and Supplementary Fig. 6a). Differences in local gene expression can be explained by either difference in cellular composition or cell-type-specific gene expression. We used cell type prediction scores for spatial transcriptomes as a proxy for cellular composition and analyzed temporal changes in local cellular composition for the major anatomical regions within ventricular tissue (Fig. 3b). We observed a decline in the proportion of the undifferentiated and immature cells and an increase in differentiated cells across stages in all ventricular regions, as expected. The average proportion of endothelial cells decreased in both ventricles, likely due to the compaction of trabecular cardiomyocytes and thickening of the compact myocardium throughout developmental time (Supplementary Fig. 6e). Two to eight-cell types contributed to each local transcriptome (number of distinct cell types with a prediction score greater than 5%) with a low local cell-type heterogeneity in the valve region and high heterogeneity in the trabecular regions of the heart (Fig. 3c). On day 14, we observed spots with high heterogeneity interspersed in the compact ventricle, which surrounds vascular networks (Fig. 3c, Supplementary Fig. 2j).

To detect region-specific markers, we performed differential gene expression analysis between anatomical regions on individual stages (Supplementary Fig. 6a). Interestingly, we found stage-dependent transcriptional differences in left and right ventricles at both day 7 and day 10. These differences diminished by day 14 when the main cardiogenesis events were complete. T-box transcription factor 5 (*TBX5*) expression was significantly enriched in the left ventricle as compared to the right and this difference decreases with a stage from days 7 to day 14 (Fig. 3d). Chromogranin B (*CHGB*) expression was mostly restricted to the right ventricle from day 7 onwards and *ACTG2* expression to the right ventricle on day 7 (Fig. 3f and 3g). *TBX5* specifies the positioning of the left and right ventricular chambers and has been shown to be enriched in the left ventricle of developing chicken hearts[34]. We note that the same study by Takeuchi et al. also reported right-ventricle specific expression of *TBX20*, but our spatial transcriptome data did not corroborate this finding (Fig. 3e). scRNA-seq datasets from day 7 to day 14 did not capture the sharp differences in *TBX5*, *ACTG2*, and *CHGB* ventricular expression, although they did seem to trend towards being expressed in a higher proportion of cells in their proposed ventricles (Supplementary Fig. 6b). smFISH confirmed the upregulation of *TBX5* in the trabecular cardiomyocytes of the left ventricle and identified that this enrichment decreases with the developmental stage (Fig. 3i). smFISH experiments further confirmed a very low but equal expression of *TBX20* in the left and right ventricles across stages (Fig. 3i). Our analysis also identified that myoglobin (*MB*) expression was spatially restricted to the developing compact myocardial layers in the later stages, indicative of cardiomyocytes transitioning to a more mature phenotype with a greater demand for oxygen (Fig. 3h). By accounting for local cell-type composition, we found that this *MB* upregulation is a result of both an increase in mature myocytes and increased *MB* expression in myocyte cells in compact myocardium (Fig. 3b, Supplementary Fig. 6c). On the contrary, the natriuretic peptide B (*NPPB*) gene was spatially

restricted to developing trabeculated myocardial layers, as expected[35–37] (Supplementary Fig. 6d).

We further explored spatial RNA-seq datasets to identify Spatio-temporal patterns in the expression of congenital heart defects (CHD) associated genes. Loss of function dominant mutations in over 50 human genes have been found to be associated with CHD. Expression of these genes occurs in highly specified temporal–spatial patterns throughout development, a level of regulation that might predict there could be strong correlations between genotype and phenotype in CHD[38–40]. We performed spatial RNA-seq analysis on genes implicated in CHD and found that the expression of crucial transcription factors like *GATA5* was enriched in the valve region, IRX4 expression was restricted to the ventricular tissue, and *TBX5* was enriched in the left ventricle (Fig. 4a–b and Fig. 3d). *PITX2* gene which is a master regulator of laterality in early plate mesoderm and a critical component in determining organ laterality was enriched in the left side of the heart as expected (Fig. 4c). Expression of CHD-associated sarcomeric protein-coding genes like *ACTC1* and *GJA5* were enriched in both atrial and ventricular tissue but *MYH7* was restricted to atrial chambers only (Fig. 4d–f).

**A persistent enrichment of thymosin beta-4 expression in coronary vascular cells.** Unsupervised clustering of single-cell transcriptomes identified a heterogeneous cell cluster enriched in *TMSB4X* containing cells from all four developmental stages (*TMSB4X* High Cells in Fig. 1b, Supplementary Fig. 7a). *TMSB4X* encodes thymosin beta-4, a well-known pleiotropic, secreted small peptide, which plays an important role in the actin-cytoskeletal organization, cellular motility, survival, and differentiation[41]. Because little is known about the spatiotemporal and cell-type-specific expression profile of thymosin beta-4 during cardiogenesis, we investigated the heterogeneity of cellular phenotypes and their transcriptional profile within the *TMSB4X* high cluster in depth.

We first re-clustered 1075 *TMSB4X* high cells from this cluster and examined its cell-type composition (Fig. 5a). We found that cluster 1 mainly consisted of epicardial cells expressing *TCF21* and endocardial cells expressing endocardial markers *IRX6*[42] and *NPC2*[43] (Fig. 5a, right). Cluster 2 mainly consisted of vascular mural cells that differentially express *ACTA2* (Fig. 5a, right). Cluster 3 mainly consisted of coronary vascular endothelial cells that differentially express *FABP5* and *TM4SF18* (Fig. 5a, right). *TMSB4X* was found to be enriched in valves and the ventricular wall at later stages in the spatial RNA-seq data (Supplementary Fig. 7b). Further analysis of scRNAseq data identified a slight increase in thymosin beta-4 expression with developmental stage and highest thymosin beta-4 expression within the coronary vascular endothelial cell subcluster 2 (Supplementary Fig. 7c). Cell type prediction scores identified that the "*TMSB4X* high cells" were detected within regions of the outermost layer of epicardium on day 4 and interspersed throughout the compact myocardium at day 14, which most likely represent coronary vascular cells (Supplementary Fig. 2k). One other beta-thymosin, *TMSB15B*, was expressed but *TMSB15B* expression did not change with the stage (Supplementary Fig. 7e).

Differential gene expression analysis of the *TMSB4X* high cell cluster identified significant upregulation of genes associated with cytoskeleton organization (vimentin (*VIM*), Rho GTPase (*RHOA*), actin beta (*ACTB*), actin gamma 1 (*ACTG1*), and Destin actin-depolymerizing factor (*DSTN*)) (Fig. 5b). In addition, we observed upregulation of calcium-binding proteins calmodulin 1 (*CALM1*) and calmodulin 2 (*CALM2*), which are associated with cell cycle progression, proliferation, and signaling and are known to be activated by thymosin beta-4[44,45]. Overall,

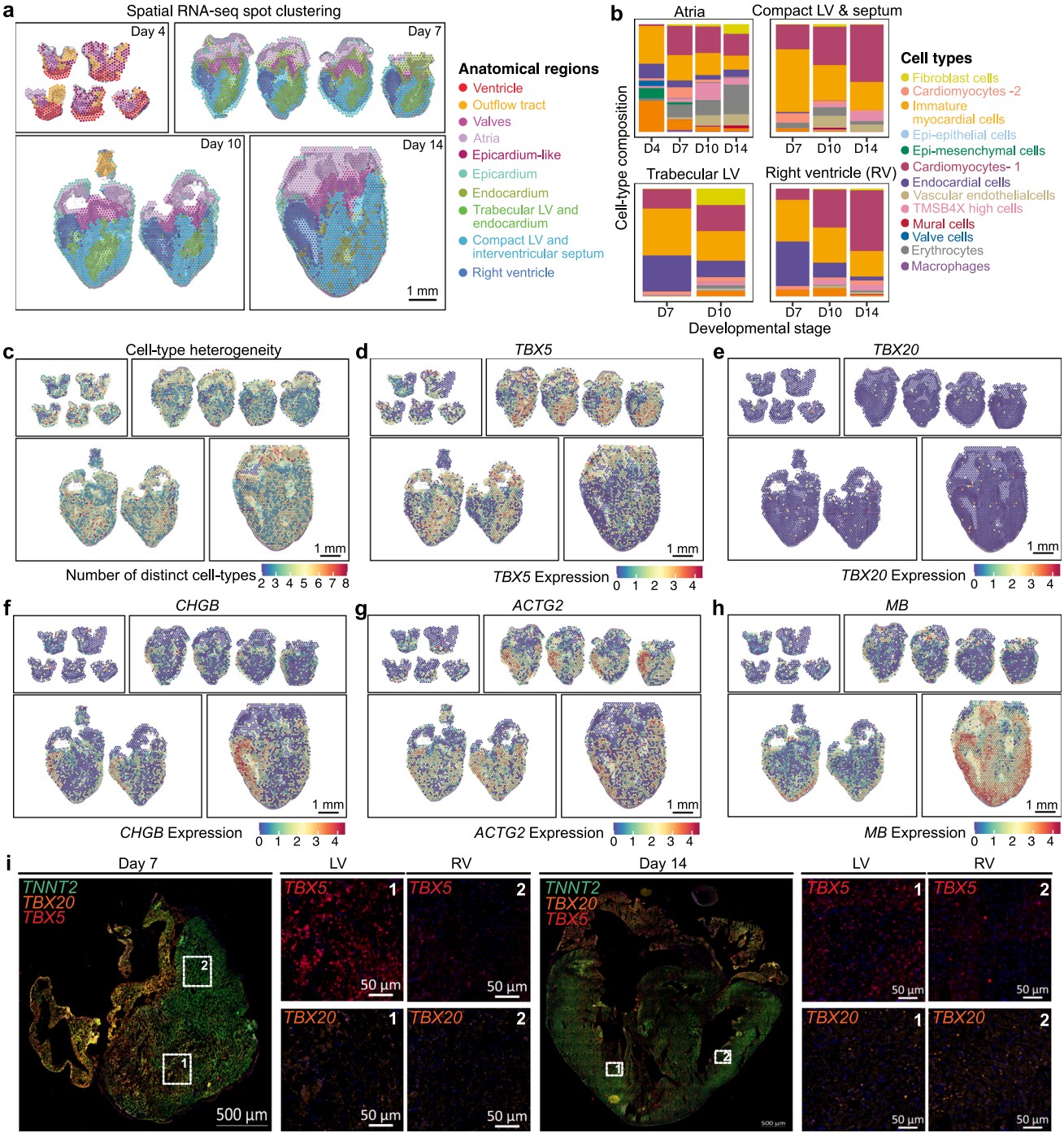

**Fig. 3 Spatial RNA-seq identifies spatially restricted genes in cardiac tissue during development. a** Spatial RNA-seq barcoded spots clustered by gene expression and labeled by tissue anatomical compartment for four developmental stages. **b** Average cell type composition across various tissue anatomical compartments. **c** Spatial map showing the cell type heterogeneity for every spot in cardiac tissue across stages. Cell type was estimated by enumerating the number of distinct cell-types with prediction scores greater than 5%. **d**–**h** Spatially resolved gene expression for spatially restricted genes. **d** *TBX5* overexpressed in left ventricles on day 7 and day 10. **e** *TBX20* showing no spatial restriction. **f** *CHGB* overexpressed in right ventricles across stages. **g** *ACTG2* overexpressed in right ventricles on day 7. **h** *MB* expressed in compact myocardium on day 10 and day 14. MB expression increases with the developmental stage. **i** smFISH stained day 7 and day 14 hearts for TBX5 (red), TBX20 (orange), *TNNT2* (green), and DAPI (blue) confirming *TBX5* overexpressed in left ventricles of day 7 hearts but not day 14 and *TBX20* showing no left or right ventricle expression differences. Representative images of three to four biological replicates.

we conclude that *TMSB4X* high cells exhibit increased cytoskeleton organization activity and calcium signaling, as confirmed by gene ontology analysis (Supplementary Fig. 7d).

To further characterize the specific cell types of these thymosin beta-4 high cells, we performed smFISH and immunohistochemistry

experiments (see "Methods" section). At the day 4 (HH24) stage, the highest thymosin beta-4 mRNA expression was in the epicardium and endocardium (Fig. 5c, Supplementary Fig. 8a and Supplementary Fig. 8b). In day 7 (HH31) hearts, thymosin beta-4 mRNA expression was highest in developing coronary vascular endothelial

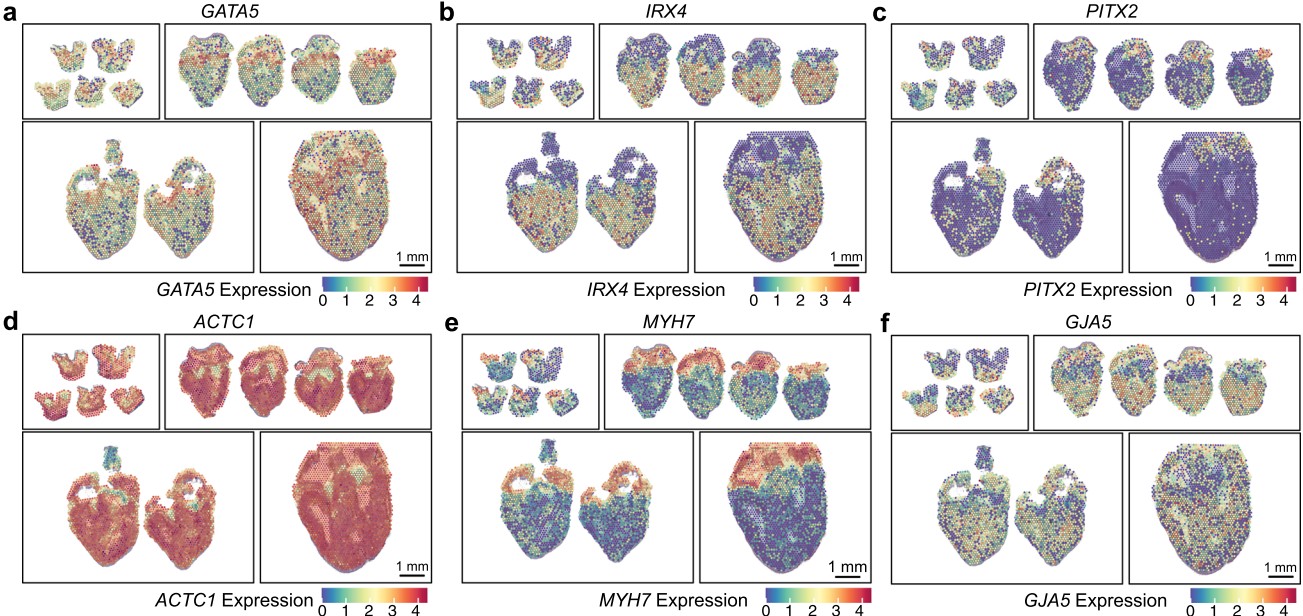

**Fig. 4 Spatially resolved gene expression for congenital heart disease (CHD) associated genes. a** GATA5, (**b**) IRX4, (**c**) PITX2, (**d**) ACTC1, (**e**) MYH7, and (**f**) GJA5.

cells as demonstrated by the overlap of the endothelial marker CDH5 with TMSB4X and developing mural cells via overlap of PDGFRB with TMSB4X (Fig. 5d, Supplementary Fig. 8a, and Supplementary Fig. 8c). By the HH36 (day 10) stage, the highest thymosin beta-4 expression was observed in CDH5+ vascular endothelial cells and surrounding mural cells (CDH5−, PDGFRB+) (Fig. 5e, Supplementary Fig. 8a, and Supplementary Fig. 8d). By the HH40 (day 14) stage, TMSB4X expression remains high in CDH5+ vascular endothelial cells and a lower expression in PDGFRB+ CDH5− mural cells (Fig. 5f, Supplementary Fig. 8a, and Supplementary Fig. 8e). Immunohistochemistry assays further corroborated these findings (Supplementary Fig. 9a–f). Together, combined spatially resolved transcriptomic and imaging data shows that TMSB4X is ubiquitously expressed by all cell types but enriched within coronary vascular cells from the initiation of coronary vasculature development through the maturation of the coronary vasculature during ventricular development.

## Discussion

We combined single-cell and spatial transcriptomics to create a hierarchical map of cellular lineages in the developing chicken heart. The dataset spans four developmental stages, 22,315 single-cell transcriptomes, and 12 spatial gene expression maps. By combining spatial and scRNA-seq assays using bioinformatic approaches, we analyzed cellular proximities within cardiac tissue, to measure changes in cellular composition with anatomical location, and to quantify anatomically restricted gene expression. Because cellular transitions in complex lineages do not occur in a synchronized manner, the data represents a broad range of cellular states even though we have investigated just four ventricular developmental stages. We mapped information about lineage differentiation transitions, obtained from pseudotime ordering of single-cell transcriptomes, on spatial maps, and were thereby able to demonstrate how cellular differentiation and morphological changes co-occur. Our analysis thus demonstrates how combined spatial and single-cell RNA-sequencing can be used to study both molecular and morphological aspects of development at high spatial and temporal resolution.

Our study provides insights into the transcriptional profiles of epicardial and epicardial-derived cells across ventricular

development. The data suggest epicardial progenitor-derived cells (EPDCs) undergo EMT into the myocardium prior to fate specification since they maintain a progenitor-like transcriptional profile, which has been corroborated by other studies[46,47]. What triggers these EPDCs to undergo EMT at specific developmental times and locations remains to be fully elucidated but our data suggests extracellular matrix cues are significantly involved in this process as we show an upregulation of many ECM factors implicated in cell migration, including AGRN, EGFL7, POSTN, and FN1. Just recently, studies have demonstrated agrin's critical role in regulating epicardial EMT[48]. Previous studies have also reported upregulation of POSTN and FN1 in EPDCs undergoing EMT in mice[46,47]. EGFL7 is a secreted growth factor known to be expressed by endothelial cells and critical to angiogenesis[18] but has not been implicated in epicardial development. Interestingly, EGFL7 transcripts have been shown to be present in a subset of murine neonatal fibroblasts that sharply declines during the murine neonatal heart regenerative window[49]. Further elucidation of the factors and programs involved in epicardial development will provide valuable insight for cardiac regenerative purposes in the adult context.

The data presented here furthermore provides much-needed clarity into the expression profile of thymosin beta-4 in specific cardiac cell types throughout ventricular development. The role of thymosin beta-4 in heart development has been under significant debate in the last decade. Cardiomyocyte-specific and endothelial-specific shRNA knockdowns of thymosin beta-4 have been shown to lead to cardiac abnormalities but no abnormalities were found for global, cardiomyocyte-specific, and endothelial-specific knockouts[50–53]. These discrepancies are thought to be explained by compensatory mechanisms triggered when thymosin beta-4 is completely ablated in the knockouts, such as the use of other prevalent beta thymosins that have similar functional activity[54,55]. Our analysis demonstrates that thymosin beta-4 is ubiquitously expressed by all cell types but remains significantly enriched at the mRNA and protein levels in only the coronary vasculature cells at all the stages analyzed. A previous study in one late fetal human heart stage demonstrated high protein expression within vascular endothelial cells only low expression in cardiomyocytes[56], in agreement with our observations. Beta

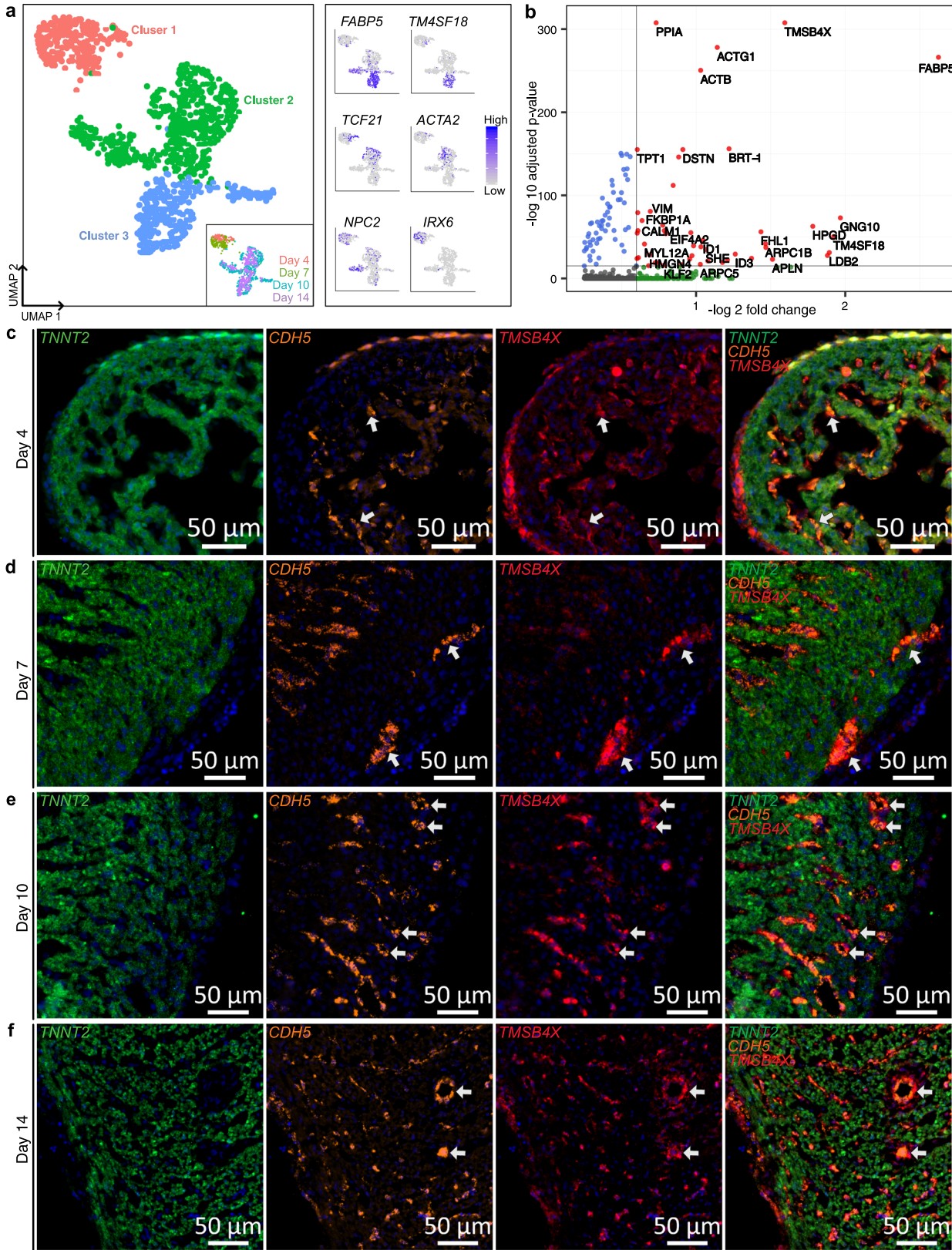

thymosins have regenerative potential as demonstrated in adult myocardial infarction models[41,57,58]. Since beta thymosins can affect many biological processes as pleiotropic secreted factors[41,57,58], a deeper understanding of how beta thymosins and corresponding compensatory mechanisms influence the biological process of specific cardiac cell types during ventricular development could lead to the discovery of treatments for myocardial infarctions and congenital cardiac diseases.

Our analysis demonstrates several ways in which spatial RNA-seq can be used to improve the robustness of scRNA-seq, and vice versa. scRNA-seq requires surgical isolation of tissues of interest, making scRNA-seq prone to contamination by cells from

**Fig. 5 A persistent enriched thymosin beta-4 expression in coronary vascular cells. a** UMAP projection of 1,075 scRNA-seq *TMSB4X* high cells clustered by gene expression (left). Inset shows the UMAP projection of *TMSB4X* high cells labeled by developmental stage. Feature plots showing expression of gene markers for cell types in the "*TMSB4X* high cells" cluster: *FABP5* and *TM4SF18* for vascular endothelial, *ACTA2* and *TCF21* for mural cells, *NPC2* and *IRX6* for endocardial cells (right). **b** Volcano plot showing differentially expressed genes for *TMSB4X* high scRNA-seq cells computed using two-sided Wilcoxon Rank-Sum test. Dotted lines represent thresholds for significantly enriched genes (red): $-\log_2$ fold change > 0.5 and *p*-value <$10^{-5}$. **c–f** smFISH images of the chicken heart ventricular free wall sections across four developmental stages labeled for cardiomyocyte cell marker *TNNT2* (green), endothelial cell marker *CDH5* (orange), thymosin beta-4 *TMSB4X* (red), and DAPI (blue). Representative images of three to four biological replicates. **c** Day 4 (HH24); Cells with high autofluorescence in all channels are Erythrocytes, (**d**) day 7 (HH31), (**e**) day 10 (HH36), (**f**) day 14 (HH40).

adjacent tissues. Matched spatial RNA-seq can be used to identify such cellular contamination. For example, using spatial RNA-seq, we identified contamination of valve cells in ventricular scRNA-seq data collected at early stages. In addition to cellular contamination, scRNA-seq is sensitive to contamination by cell-free RNA released from cells that lyse during tissue dissociation, which is not the case for spatial RNA-seq. scRNA-seq data for matched tissue can be used to deconvolve spatial RNA-seq transcriptome to predict cell type compositions. These cell type predictions allow one to analyze and compare the relative composition of cell types per spot within and between tissues, albeit only for sections analyzed.

We note that previous scRNA-seq studies have studied the development of human and mouse hearts[33,35,59]. These studies implemented low-throughput scRNA-seq methodologies and consequently suffered from limitations in cell-type resolution. A very recent study combined spatial transcriptomics combined with scRNA-seq to construct a spatiotemporal organ-wide gene expression and cell atlas of developing human hearts at three developmental stages in the first trimester: 4.5–5 (≈HH28-31), 6.5 (≈HH34-37), and 9 post-conception weeks (≈HH40-later)[60]. However, the study has limitations with respect to detecting rare cell types and identifying spatiotemporal cell-cell interactions due to the limited cell types being detected (total of only 3717 cells), limited genes being used in the in-situ sequencing (ISS) panel, and low resolution of the spatial transcriptomic technique (3115 spots containing ~30 cells per spot). Therefore, we were able to probe earlier stages and a broader range of stages of cardiac development, capture greater cell phenotypic heterogeneity, and show several aspects of epicardial and vascular development.

In conclusion, we have combined single-cell and spatial transcriptomics to explore the early development of chicken hearts at high molecular, spatial, and temporal resolution. We constructed a single cell, spatially resolved gene expression census, uncovered several regulatory mechanisms, and demonstrated that spatiotemporal single-cell RNA sequencing can be used to study the interplay between cellular differentiation and morphogenesis.

## Methods
**Sample preparation for single-cell transcriptomics.** We used fertile bovans brown chicken (*Gallus gallus*) eggs to study the development of the heart on four key embryonic stages. Eggs were obtained from a farmer and were then incubated in a standard humidity and temperature-regulated egg incubator until the embryonic day of interest. The use of chick embryos at these developmental stages does not require ethical board approval. All embryos were quickly isolated and euthanized via decapitation. Ventricles were isolated aseptically, placed in ice-cold Hank's Balanced Salt Solution, and minced into 1–2 mm pieces. Six dozen day 4 (HH21-24) whole ventricles, four dozen day 7 (HH30-31) left and right ventricles, three dozen day 10 (HH35-36) left and right ventricles, and one dozen day 14 (HH40) left and right ventricles were pooled, respectively, for a total of seven samples to be analyzed via single-cell RNA sequencing. Ideally, scRNA-seq requires the tissue to be collected and used right away to not lose expression and get good cell viability. After a lot of practice on isolating hearts from early stages, we were able to isolate several dozen hearts in around an hour which was placed in Hank's Balanced Salt Solution on ice the whole time to minimize changes in expression in cells. Day 4 (HH21-24) and day 7 (HH30-31) ventricular samples were digested in 1.5 mg/mL collagenase type II/ dispase (Roche) for one cycle of 20 min and one

cycle of 10 min under mild agitation at 37 °C. Day 10 (HH35-36) and day 14 (HH40) ventricular samples were digested in 300 U/mL collagenase type II (Worthington Biochemical Corporation) for four cycles of 10 min under mild agitation at 37 °C. At the end of the digestions, the cells were passed through a 40 μm filter and centrifuged into a pellet. To remove most blood contaminants, samples were resuspended in an ammonium-chloride-potassium red blood cell lysis buffer (Thermo Fisher Scientific) and centrifuged again. Samples were then resuspended in phosphate-buffered saline with 0.04% bovine serum albumin (Thermo Fisher Scientific) at $1 \times 10^6$ cells per mL. Cells from each sample were stained with Trypan Blue and cell viability was calculated on automated cell counters before loading the cells on 10x Chromium. The samples had a cell viability of around ~87–89% for early stages and 92–95% for later stages. Lower cell viability in earlier stages could be explained by the delay in tissue processing due to pooling dozens of hearts in a sample. We used these cell viabilities to adjust the number of cells loaded on 10x Chromium to get the desired number of transcriptomes from viable cells.

**Single-cell RNA sequencing library preparation.** 4000–5000 cells per sample (for day 4 and day 7 samples) and 2000–3000 cells per sample (for day 10 and day 14 samples) were targeted on the Chromium platform (10x Genomics) using one lane per time point. Single-cell mRNA libraries were built using the Chromium Next GEM Single Cell 3' Library Construction V3 Kit for day 4 and day 7 samples and Chromium Next GEM Single Cell 3' Library Construction V2 Kit for day 10 and day 14 samples, libraries sequenced on an Illumina NextSeq 500/550 using 75 cycle high output kits (Index 1 = 8, Read 1 = 28, and Read 2 = 55) for day 4 and day 7, 75 cycle high output kits (Index 1 = 8, Read 1 = 26, and Read 2 = 98) for day 10, and 75 cycle high output kits (Index 1 = 8, Read 1 = 26, and Read 2 = 58) for day 14. We observe a sequencing saturation of ~31% for day 4, ~50% for day 7 ventricles, ~91% for day 10 ventricles, and 94% and ~78% for day 14 ventricles. Sequencing data were aligned to chicken reference (assembly: GRCg6a) using the Cell Ranger 3.0.2 pipeline (10x Genomics).

**Reference genome and annotation.** *Gallus gallus* genome and gene annotations (assembly: GRCg6a) were downloaded from Ensembl genome browser 97 and processed using the Cell Ranger 3.0.2 (10x Genomics) pipeline's mkref command. The reference was then used in the Cell Ranger "count" command to generate expression matrices.

**scRNA-seq data processing, batch correction, clustering, cell-type labeling, and data visualization.** After excluding the cells with less than 200 unique genes or more than 20 percent mitochondrial transcripts, we analyzed 5653, 8463, 5190, and 3009 single-cell transcriptomes from day 4, day 7, day 10, and day 14, respectively. The 22,315 cells from 4 stages were transformed, normalized, and scaled using the Seurat V3 package, and then used for batch correction. During log-normalization, feature counts for each cell are divided by the total counts for that cell, then multiplied by the scale.factor ($10^6$), and then natural-log transformed. Scanorama[4] was used for dataset integration and batch correction using transformed and normalized expression values. We used the batch corrected values for further processing and analysis. The seurat-v3 package was used to select top variable genes for scRNA-seq clustering. We used the FindVariableFeatures function in Seurat to choose the top 2000 highly variable genes from the dataset using the "vst" selection method. We then performed mean centering and scaling, followed by principal component analysis (PCA) on a matrix composed of cells and batch-corrected scanorama-output values, and reduced the dimensions of the data to the top 20 principal components. Uniform Manifold Approximation and Projection (UMAP) was initialized in this PCA space to visualize the data on reduced UMAP dimensions. The cells were clustered on PCA space using the Lovain algorithm on k-nearest neighbors graph constructed using gene expression data as implemented in FindNeighbors and FindClusters commands in Seurat-v3. The method returned 17 cell clusters which were then visualized on UMAP space using the DimPlot command. Cell-type-specific canonical gene markers were used to label clusters differentially expressing those markers. To accurately label individual clusters, the Wilcox test was performed to find differentially expressed genes for each cluster. We used the FindAllMarkers function in Seurat to get a list of differentially expressed genes for each cluster. Gene expression was visualized using FeaturePlot, DoHeatMap, and VlnPlot functions from Seurat-v3. Cells were

grouped into lineages using gene markers and then used for trajectory construction and pseudotime analysis.

**Sample preparation for 10x Visium spatial transcriptomics platform**. Whole hearts were isolated using aseptic technique and placed in ice-cold sterile Hank's Balanced Salt Solution and then blood was carefully removed by perfusing the hearts through the apex. Fresh tissues were immediately embedded in Optimal Cutting Compound (OCT) media and frozen in liquid-nitrogen-cooled isopentane bath, cut into 10 μm sections using Thermo Scientific CryoStar NX50 cryostat, and mounted on 10x Visium slides, which were pre-cooled to −20 °C.

**10x Visium spatial transcriptomics library preparation**. We used the 10x Genomics Spatial RNAseq Visium platform for the spatial transcriptomics experiments. A 10x Genomics Visium Gene Expression slide has 4 capture areas each with an array of 5000 circular spots containing printed DNA oligos for mRNA capture. These oligos on each spot have a PCR handle, unique spatial barcode, Unique Molecular Identifier (UMI), and a poly-dT-VN tail for capturing the 3' end of mRNA molecules. Each spot with a unique spatial barcode is 55 μm in diameter and the center to center distance between the spots is 110 μm. One 55 μm spot captures mRNA from 10 to 20 cells depending on cell size and packing density which is variable across the tissue. Tissue sections from fresh frozen chicken embryonic hearts were mounted for 4 stages (five sections from day 4-HH24, four sections for day 7-HH31, two sections for day 10-HH36, and one section for day 14-HH40) with one stage per capture area on a 10x Visium gene expression slide. The sample for day 4 had three biological replicates with two of them having technical replicates and the samples for day 7 and day 10 had four and two biological replicates, respectively. These sections are then fixed in pre-chilled methanol for 30 min and then H&E stained and imaged. H&E staining is later used by the 10x Genomics Cell Ranger software to detect the spots which are covered by the tissue. Optimal permeabilization time for 10 μm thick chicken heart sections was found to be 12 min using 10x Genomics Visium Tissue Optimization Kit. Spatially tagged cDNA libraries were built using the 10x Genomics Visium Spatial Gene Expression 3' Library Construction V1 Kit. H&E stained heart tissue sections were imaged using Zeiss PALM MicroBeam laser capture microdissection system and the images were stitched and processed using Fiji ImageJ software. cDNA libraries were sequenced on an Illumina NextSeq 500/550 using 150 cycle high output kits (Read 1 = 28, Read 2 = 120, Index 1 = 10, and Index 2 = 10). Fluidigm frames around the capture area on the Visium slide were aligned manually and spots covering the tissue were selected using Loop Browser 4.0.0 software (10x Genomics). Overall, tissue sections from day 4 to day 14 covered a total of 747, 1966, 1916, and 1967 spots on the capture area, respectively. Sequencing data were then aligned to the chicken reference genome using the Space Ranger 1.0.0 pipeline to derive a feature spot-barcode expression matrix (10x Genomics).

**Spatial RNA-seq data processing, integration, and visualization**. Spatial RNAseq data from 6800 barcoded spatial spots from four 10x Visium capture areas was log-normalized using the Seurat V3.2 package and then used for batch correction. During log-normalization, feature counts for each spot are divided by the total counts for that cell, then multiplied by the scale.factor ($10^6$), and then natural-log transformed. Seurat-v3.2 package was then used to select top variable genes for spatial RNA-seq clustering. We used the FindVariableFeatures function in Seurat to choose the top 2000 highly variable genes from the dataset using the "vst" selection method. We then performed mean centering and scaling, followed by principal component analysis (PCA) on a matrix composed of spots and gene expression (UMI) counts, and reduced the dimensions of the data to the top 20 principal components. Uniform Manifold Approximation and Projection (UMAP) was initialized in this PCA space to visualize the data on reduced UMAP dimensions. The spots were clustered on PCA space using the Shared Nearest Neighbor (SNN) algorithm implemented as FindNeighbors and FindClusters in Seurat v3.2 with parameters $k = 30$, and resolution = 0.5. The method returned spot clusters representing anatomical regions in the tissues, which were then visualized on UMAP space using the SpatialDimPlot command. To accurately label anatomical regions, the Wilcox test was performed to find differentially expressed genes for each region. We used the FindAllMarkers function in Seurat with its default parameters to get a list of differentially expressed genes for each cluster. Gene expression was visualized using the SpatialFeaturePlot function from Seurat v3.2. An anchor-based integration method implemented in Seurat-v3.2 was used for integration of spatial RNA-seq data with time-matched scRNA-seq data using FindIntegrationAnchors command and then cell type labels were transferred to spatial data using TransferData command. Cell type prediction values for Spatial RNA-seq spots were saved as an assay and used for further analysis. Cell type colocalization values were calculated by counting cell type pair abundances in spatial RNA-seq spots. Only cell types with the top four prediction scores in each spot were included. We also constructed a cell-spot similarity map by transferring cell barcode IDs to spatial barcoded spots. The cell-spot similarity matrix containing scRNA-seq cell similarity prediction for each spot in scRNA-seq data, which we further used to estimate pseudotime for spatial RNA-seq spots.

**Pseudotime analysis and trajectory construction**. We used PHATE[7] (Potential of Heat-diffusion for Affinity-based Transition Embedding) to visualize developmental trajectories because of its ability to learn and maintain local and global distances in low dimensional space. We reclustered cells from individual lineages, performed PHATE dimension reduction on scanorama integrated values, and used PHATE1 dimension as a proxy for development time. PHATE reduction was performed using the phate command implemented in the R package: phateR. We also used monocle-v2[20,21] to order the cells in epicardial, endocardial, and myocardial lineages along pseudotime and reconstruct lineage trajectories. We filtered the genes detected in the dataset and retained the top 2000 highly variable genes calculated using Seurat-v3 in our monocle analysis. We further filtered these genes to genes differentially expressed in cell type subclusters within the lineage using differentialGeneTest command and then reduced the dimension of the data using the DDRTree method. We used the ReduceDimension function in monocle-v2 to reduce the dimension to two DDRTree components, which is then used to define a pseudotime scale. The cells were then ordered along pseudotime using monocle's orderCells command, and the root of the trees was defined as the branch with maximum cells from Day 4 samples. The gene expression changes along pseudotime based trees were then visualized using PseudotimeHeatMap command in monocle-v2. To estimate pseudotime for spatial maps, we used the spot-cell similarity matrix and estimated pseudotime for individual lineages in spatial RNA-seq data. We defined spot pseudotime for a lineage as the average of scRNA-seq pseudotime (PHATE1 dimension or monocle pseudotime) for cells having a non-zero similarity prediction with that spot. This spot pseudotime was then visualized on spatial maps using the SpatialFeaturePlot command in Seurat-v3.2.

**Enrichment analysis for Gene Ontology (GO)**. We used the genes differentially expressed between clusters in individual lineages to perform enrichment analysis for Gene Ontology. We selected significant genes using their p-value scores from differential expression test and used the classic fisher test implemented in topGO[61] R package for enrichment analysis of GO terms representing various Biological Processes (BP).

**smFISH split probe design strategy and Signal Amplification using HCR v3.0**. A two-step hybridization strategy with split probe design and Hybridization Chain Reaction HCR v3.0[62] was used to label up to three transcripts in a single tissue section. Probes were designed using NCBI primer-blast which uses primer3 for designing internal hybridization oligo and BLASTn to check for binding specificity. We designed 19–21 bp primer pairs for an amplicon length of 38–42 bp, primer melting temperature between 57 °C and 63 °C, and primer GC content between 35% and 70%. 7–13 sets of reverse complemented forward primers and reverse primers were then concatenated to flanking sequencing for HCR, ordered from IDT for each gene target, mixed together, and diluted in nuclease-free water to create a split probe pool stock solution at 10 μM for every target. The exact sequences of all probes used for the smFISH experiment have been provided in Supplementary Data 3. Hybridization Buffer (HB) was prepared with 2× SSC, 5× of Denhart Solution, 10% Ethylene Carbonate, 10% Dextran Sulfate, 0.01% SDS, 1uM of probe pool mix for the hybridization reaction. Hairpin pairs labeled with three different fluorophores namely Alexa-488, Alexa-546, and Alexa-647 (Molecular Instruments, Supplementary Data 4) were used for hybridization Chain Reaction v3.0. Amplification Buffer (AB) was prepared with 2× SSC, 5× of Denhart Solution, 10% Dextran Sulfate, 0.01% SDS, 0.06 μM of HCR hairpins for the amplification reaction. Two microliters of each fluorophore-labeled hairpins at 3 μM corresponding to the target genes were mixed, incubated at 95 °C for 1.5 min, covered in aluminum foil, and left to cool down at room temperature for 30 min to form hairpins before adding it to Amplification Buffer (AB).

**Multiplexed smFISH assays**. Whole hearts were isolated using aseptic technique and placed in ice-cold sterile Hank's Balanced Salt Solution and then blood was carefully removed by perfusing the hearts through the apex. For day 7 (HH30), day 10 (HH36), and day 14 (HH40), fresh tissues were immediately embedded in Optimal Cutting Compound (OCT) media and frozen in liquid nitrogen-cooled isopentane, cut into 10 μm sections using a Thermo Scientific CryoStar NX50 cryostat, and mounted on −20 °C cooled histological glass slides which were then stored at −80 °C until used. For the day 4 (HH24) stage, whole embryos were isolated and fixed in 4% paraformaldehyde for about 16–24 h at 4 °C. Slides with tissue sections were then brought to room temperature until the sections melted and were then immediately fixed in 4% paraformaldehyde for 12 min. Post fixation, the sections were washed for 5 mins in 1× PBS twice, incubated for 1 h in 70% ethanol for tissue permeabilization, washed again for 5 mins in 1× PBS, and then used for primary hybridization. Twenty microliters hybridization mix (with probes) was then put on each slide to cover the tissue section placed inside a Biorad Frame Seal 9 mm × 9 mm chambers and incubated overnight at 37 °C inside a humidifying chamber. After primary hybridization, frame seals were removed and slides were washed in Hybridization Wash Buffer (0.215 M NaCl, 0.02 M Tris HCl pH 7.5, and 0.005 M EDTA) for 20–30 min at 48 °C. Twenty microliters amplification mix was then put on each slide to cover the tissue section placed inside a Biorad Frame Seal 9 mm × 9 mm chambers and incubated for 6–8 h at room temperature in dark for signal amplification. After signal amplification, frame seals

were removed, and slides were washed again in the wash buffer for 30–40 min at 48 °C. Last, tissue sections were then counterstained with DAPI for 10 min at room temperature, washed for 5 min in 1× PBS, excess PBS cleaned using Kimwipe, immediately mounted on coverslips using Prolong glass antifade media, left overnight for treatment, and imaged the next day on a Zeiss Axio Observer Z1 Microscope using a Hamamatsu ORCA Fusion Gen III Scientific CMOS camera. smFISH images were processed and thresholded on Zen 3.1 software (Blue edition).

**Immunohistochemistry assays**. Whole hearts were isolated using aseptic technique and placed in ice-cold sterile Hank's Balanced Salt Solution and then blood was carefully removed by perfusing the hearts through the apex. For day 7 (HH30), day 10 (HH36), and day 14 (HH40) heart samples after perfusion, tissues were fixed in 4% paraformaldehyde for about 16–24 h at 4 °C. For the day 4 (HH24) stage, whole embryos were isolated and fixed in 4% paraformaldehyde for about 16–24 h at 4 °C. Samples were then processed, embedded in paraffin, cut into 6 μm sections using a microtome, and mounted onto histological glass slides. Slides were incubated for 20 min at 58 °C to melt paraffin, washed three times in xylene for 3 min each, and then placed in decreasing ethanol concentrations to rehydrate slides. Samples then underwent an antigen retrieval step via incubation in 1× citrate buffer for at least 10 min at 95 °C. Samples were then permeabilized in 0.3% Triton X-100 in tris buffered saline for 15 min, washed three times in 0.05% Tween-20 in tris buffered saline (TBST), blocked for 1 h at room temperature in blocking buffer (1% bovine serum albumin and 5% goat serum in TBST), washed once in TBST, and incubated in primary antibodies diluted in antibody solution (1% bovine serum albumin in TBST) overnight at 4 °C. Primary antibodies used were mouse anti-MF20 antibody (1:100, 14650382, Invitrogen), rabbit anti-TMSB4X antibody (1:200, ab14335, Abcam), rabbit anti-CDH5 antibody (1:200, ab33168 Abcam), rabbit anti-ACTA2 antibody (1:200, ab5694, Abcam). After overnight primary incubation, samples were washed three times in TBST and then incubated in secondary antibodies for 1 h at room temperature. The secondary antibodies were goat anti-mouse 488 (1:500, A11017, Invitrogen) and goat anti-rabbit 555 (1:500, A21430, Invitrogen). Lastly, samples were washed in TBST, stained with DAPI (1:1000, Thermofisher), and mounted. Images were acquired on a Zeiss Axio Observer Z1 Microscope using a Hamamatsu ORCA Fusion Gen III Scientific CMOS camera. Immunostaining images were processed and thresholded on Zen 3.1 software (Blue edition).

**Reporting summary**. Further information on research design is available in the Nature Research Reporting Summary linked to this article.

## Data availability
The authors declare that all data supporting the findings of this study are available within the article and its supplementary information files or from the corresponding author upon reasonable request. The authors declare that all sequencing data supporting the findings of this study have been deposited in NCBI's Gene Expression Omnibus[63] GEO database under the accession code GSE149457. H&E stained tissue images matched to spatial RNA-seq datasets and analysis scripts have been made available on GitHub repository (https://github.com/madhavmantri/chicken_heart) with the identifier (https://doi.org/10.5281/zenodo.4517120)[64].

## Code availability
The data analysis scripts have been made available on the GitHub repository (https://github.com/madhavmantri/chicken_heart) with the identifier (https://doi.org/10.5281/zenodo.4517120)[64].

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

## Acknowledgements
We would like to thank Peter Schweitzer and the Cornell Genomics Center for help with single-cell and spatial sequencing assays, the Cornell Bioinformatics facility for assistance with bioinformatics, and the Cornell Imaging facility for assistance with imaging assays. We also thank the members of the Butcher and De Vlaminck labs for valuable discussion. This work was supported by R01HL143247 (to J.T.B. and I.D.V.), R33CA235302 (to I.D.V.), R21AI133331 (to I.D.V.), and DP2AI138242 (to I.D.V.).

## Author contributions
M.M., G.J.S., J.T.B., and I.D.V. designed the study. M.M., G.J.S., and R.A. performed scRNAseq experiments. M.M. and G.J.S. performed spatial RNAseq experiments. M.M., G.J.S., H.S., and B.G. performed smFISH and immunostaining experiments. M.M., G.J.S., M.F.Z.W., and D.M. analyzed the data. M.M., G.J.S., J.T.B., and I.D.V. wrote the manuscript. All authors provided feedback and comments.

## Competing interests
The authors declare no competing interests.
