## [Peer Review File · Nature Communications]

Reviewers' Comments:

Reviewer #1:

Remarks to the Author:

This is a very interesting paper that uses the chick model of heart development to analyze spatiotemporal gene expression in the heart over developmental stages. Developmental time points chosen range from tubular heart stages to the fully formed heart. The four time points chosen are adequate.

In order to get both spatial information and single cell data, the authors use a combination of spatial RNAseq and single cell RNAseq (scRNAseq). Spatial RNAseq localizes expression to spots, but loses single cell resolution; scRNAseq has single cell resolution, but loses localization. From a theoretical perspective the project makes sense. Spatial spots are analyzed with RNAseq, and, given the gene expression and location it should be possible to infer the cell composition of the spot, and from there, one can use information from scRNAseq to complete mappings of expression and cell interactions. This is a very exciting project. However, while the idea is sound, it has not been validated. Why not using FISH or a similar technique to validate some of the findings? With FISH it is possible to get single cell resolution of gene expression and thus predicted clusters of cells, including their location and maturation, could be verified. This validation will really strengthen the paper.

Also, it would be nice to verify some findings against previous work in the literature (avian or mouse). Many of the developmental/gene expression processes (including cell differentiation and migration) that take place during heart development are already known and well studied. Do the findings fit well into known facts of heart development? This comparison and discussion will further strengthen the paper.

Some minor comments:

1. I had a difficult time understanding exactly how does the spatial RNAseq system works (e.g. how big are the spots? how many cells is it sampling? how is the 'material' collected/processed? why the preparation involves H&E staining? are there issues with processing, and how were they minimized?).
2. For scRNAseq collection, several hearts are pooled together. How exactly was this done to minimize changes in expression in cells? Typically the tissue has to be collected and analyzed right away not to lose expression. But collecting a dozen or more hearts takes time. I did not see a discussion on this or possible effects of collection methods.
3. There are no 'fetal' stages in avian - those would be embryonic stages instead. However, you can say it is a fetal-like stage.
4. Lines 154-156. When I first read it, it seemed to me that 'epicardial lineage', which is all over the heart, meant cells of epicardial origin (in general), which abound in the heart. So the data made sense. However, later on it is stated that epicardial lineage is with respect to the tubular heart stage (HH24) - so cells derived from epicardial cells at HH24. Which one is right? I don't think that you will find epicardial cells (derived from the HH24 epicardium) in the septum of the heart later on... But of course you will find neural-crest cell derived cells, which are of epicardial origin. Please clarify.
5. Could you verify the statement on lines 154-156 with previous literature findings?
6. I am not sure I understand what the pseudo-time means in the context of the paper. Could you please clarify?

7. Lines 190 to 196. Could you independently validate this finding? It should be relatively straight forward to mark the epithelial cells at HH24 and then follow them to see if they behave as described. Perhaps another group has already done similar work on this too.
8. Lines 253-3255 - proportion of endothelial cells decrease. Also due to thickening of the compact myocardium? while the endocardium continues to be a monolayer.
9. Lines 286-287 Natriuretic peptide A (NPPB)? Is this correct?
10. The coronary vasculature is of epicardial origin. Does your data fully reflect this fact? I have not seen a discussion on this, although the data seems to make sense from this point of view.
11. There is some validation performed for the TMSB4X finding. Why immunohistochemistry rather than FISH?
12. Lines 438-444: You may be reading too much into these data... it may just be needed for migration of cells to form different parts...
13. Lines 463-465: Ratios of cells cannot be derived from scRNAseq as the 'capture' efficiency of cells depends on cell type...
14. Lines 469-484: It seems that very similar studies have already been performed with human tissues. So what exactly is the novelty of your studies - in terms of technique - should be addressed, and the methods validated.

Reviewer #2:

Remarks to the Author:

The authors use scRNAseq and spatial transcriptomic technologies to evaluate changes in cellular composition and spatial restriction of gene expression during cardiac morphogenesis in chick heart. From spatially resolved maps of gene expression, they infer interactions between different cell types and differentiation trajectories across developmental stages. Despite generating a valuable resource, the authors provide very little experimental validation to support their claims and the study offers limited novel insight into biological processes or regulatory mechanisms.

Interpretation of the data is heavily dependent on the sequencing quality, integration and trajectory tools used in their analysis pipeline. Thus, the predictions generated by these approaches must be rigorously tested experimentally. As well as confirming putative expression patterns, ideally at the protein level, genetic lineage tracing and perturbation experiments are required to corroborate and challenge the proposed trajectories.

Major concerns

- Spatially resolved gene expression requires validation on histological sections, of mRNA or ideally protein, co-localisation with cell-type specific markers to support the main findings e.g. expression distribution between left vs right ventricle; compact vs trabeculated myocardium. In particular, where data are potentially novel (e.g. C3 cardiomyocyte precursor-like cells) and where data appear to contradict previous findings (e.g. TBX20), experimental validation by multiple approaches is particularly important.
- Inferred trajectories denote transcriptional similarity, not necessarily bona fide biological processes. The authors should, therefore, validate the conclusions of their trajectory analysis, using immunohistochemistry/in situ hybridization, lineage tracing and genetic perturbation experiments. That said, in support of their conclusions, most of the trajectories described have been previously reported and are unfortunately not novel.
- How were epicardial progenitor cells defined? Tcf21 alone is insufficient to identify epicardium and epicardium-derived cells in the absence of lineage tracing tools. Epicardial-derived mural cells do not express Tcf21 and, moreover, non-epicardial derivatives e.g. endocardium-derived fibroblasts

also express Tcf21. Thus, trajectory analysis will be confounded by omission of cells that have lost the precursor marker upon differentiation.

- As an example of incorrect conclusion based on inappropriate methodology, the authors conclude that the endocardium undergoes a maturation process, rather than differentiating into other cell types. This is at odds with lineage tracing studies which demonstrate that the endocardium can adopt multiple fates (endothelial, cardiac/valve fibroblasts and mural cells) but can be explained by the fact that authors used only the endocardial cluster in their trajectory analysis (Supp Fig. 1F).
- I don't understand why the authors excluded vascular endothelial cells from trajectory analysis "since their lineage origin remains debated". Firstly, there is now a reasonable consensus view that these cells derive from sinus venosus and endocardium (in chick and mouse). Secondly, if the authors are confident in the power of their data to resolve differentiation trajectories, they have an ideal opportunity to evaluate possible contributions from other lineages (e.g. epicardium).
- In the absence of experimental validation, statements such as "and were thereby able to reveal how cellular differentiation and morphological changes co-occur" and "suggests its potential morphogenetic role in directing epicardial and endocardial-derived cell migration into the myocardium to initiate and maintain ventricular compaction and maturation." and (in relation to thymosin beta-4) "our analysis clarified its role in ventricular compaction and maturation" are entirely unjustified. Mechanistic experiments are needed to support these claims.
- A number of relevant studies have not been cited and discussed. As examples, left vs right ventricle and compact versus trabecular myocardial markers in relation to e.g. Li et al 2019, Yvanka de Soysa et al 2019, Li et al 2016; epicardial trajectory model should be discussed in light of Xiao et al 2018 and Lupu et al (2020).

Minor concerns

- Fig1B: There are fewer CMs/endocardial cells at D14 samples, is this down to low cell and sequencing depth?
- Fig S1A: Almost 10-fold fewer transcripts per cell at d14, compared with d7, which may result in sequencing saturation. Moreover >2000 genes/cell is required for reliable GO analysis and inferences re: biological processes.
- Total of 22,315 cells sequenced. The authors need to detail exact cell numbers per stage in the methods.
- After scanorama batch correction and integration, is there evidence of over-correction?
- The authors should present heatmaps with the top 5 or top 10 differentially expressed genes for each cluster, to allow confirmation (by readers) on appropriateness of cluster annotation. The markers highlighted are not necessarily the better characterised/ more selective. Why not e.g. Npr3/Nfatc1 for endocardium; Wt1/Upk3b for epicardium?
- Fig2B and sup. Fig3A: the authors should discuss the differences between PHATE and Monocle 2 trajectory inferences for epicardial progenitor cells.
- Figure 3B: Is cell-type composition based on the sections shown? Multiple independent samples and sections from different regions are required for this to be accurate and validation is required. The D14 data derive from one section, which cannot be representative of the whole heart, given the dorsal-ventral variation in cellular composition. The differing numbers of sections per stage will affect comparison between stages. This limitation needs to be acknowledged.
- If the TMSB4X high cell cluster comprises a heterogeneous population of cells, it is difficult to comprehend why they cluster together, rather than with the constituent cell populations. Do they still cluster if embryonic stages are analyzed separately? The authors should demonstrate TMSB4X expression levels across clusters at separate stages. It is unclear TMSB4X expression is consistently high across all stages or primarily contributed by increased expression at D14 (FigS5B).
- Macrophages and dendritic cells are described as "circulating". Can the authors exclude that some may be cardiac-resident?
- "Smooth muscle cell" cluster should be described as mural cells as the majority at these stages will be pericytes and immature smooth muscle progenitors.
- Nppb should be defined as natriuretic peptide B, not A (presumably, in trabeculae, as opposed to Nppa, which is natriuretic peptide B).

- Figure 3B: The key does not indicate the cell type label represented by grey bars.
- Figure 4A: Typo Cluser 1 should be Cluster

Reviewer #3:

Remarks to the Author:

The authors have reported a detailed atlas of the diverse cellular lineages in the developing chicken heart and revealed their spatial organization and interactions during cardiac development by using new approaches to combine high-throughput single-cell RNA-sequencing (scRNA-seq) with spatially resolved RNA-seq. Through novel anchor-based bioinformatic approaches, they delineated molecular and morphological aspects of cardiac development, including dynamic cellular interactions within heart tissue, changes in cellular composition and anatomically restricted gene expression at high spatiotemporal resolution. Finally, they found a stage-dependent role of thymosin beta 4 in the coordination of multi-lineage cellular populations for ventricular development, i.e., compaction and maturation. Although scRNA-seq has been used to date as an excellent tool to identify cellular hierarchies and molecular signatures in developmental biology (and other settings as well), it is a caveat that it does not preserve spatial information about cellular interactions and tissue morphology. Taken together, the approaches to integrate spatial and single-cell transcriptomic data, conducted in this paper, likely hold promise to identify key stage- and anatomical location-specific machinery programs that regulate organ development in embryogenesis, as their analysis appears to provide novel insights into several regulatory programs that guide ventricular development and morphogenesis. Overall, the research and manuscript described here were well designed and described; however, the several critical issues which need to be addressed should be pointed as follows.

1) The UMAP projection of 22,315 scRNA-seq of fetal chicken hearts at four developmental stages revealed 15 distinct cell type clusters (Fig. 1B), which included epi-, endo-, and myocardial developmental lineages. However, to this reviewer, it is a bit surprising that the 15 clusters did not exhibit clearly another important anatomical differences, i.e., atria and ventricles, unlike the previous similar papers studying scRNA-seq in embryonic cardiogenesis (Dev Cell. 2019, 48:475; Cell. 2019, 179:1647). Could the authors find any differences of gene expression patterns between atria and ventricular cardiomyocytes (CMs) or do sub-clustering of them among the myocardial lineage (Supple. Fig. 1F)?

2) Another concern in the UMAP projection of 22,315 scRNA-seq (Fig. 1B) is that the 15 clusters included some odd one, i.e., cell population expressing high levels of thymosin beta-4 (TMSB4X), which consisted of 3 sub-clusters including CMs, endo- and epi-cardial cells, and vascular smooth muscle- and endothelial-like cells (Fig. 4A). What's the rationale about grouping the "TMSB4X" cluster together with other definitive 14 cell-types in the first UMAP projection (Fig. 1B)? It might provide readers with a bit confusion.

3) This type of a resource paper featuring scRNA-seq data analysis is often hampered by its descriptive nature with less informative discoveries that significantly advance our notions and understandings in developmental biology. One way to resolve this weak point is to investigate several disease-caused genes' distribution and expression patterns in the experimental dataset. In this case, it is highly recommended to examine how human congenital heart disease (CHD) genes were expressed in the spatial and single-cell RNA-seq dataset reported here and to delineate some relationship, if any, between some CHD genes and spatiotemporally specific cell-types.

4) In Fig. 2E (right), there seemed like no clear differences about the endocardial or endothelial cell markers' expression among the three clusters (C1, C2, and C3) of the endocardial lineage. Is this correct? If the authors could find any stage (e.g., early C1 or late C3)-specific expression of such genes, it should be shown.

5) How were other endocardium-specific markers, such as NPR3 and NFATC1 (Circ Res. 2016, 118:1880) expressed in the endocardial lineage analyzed here?

6) In Fig. 2H showing the myocardial lineage, the 3rd cluster (C3) was not clearly defined. This cluster was very heterogeneous and seemed to consist of cells from not only day 4 and day 7 but day 10 and day 14. Thus, the terminology, "a cardiomyocyte precursor-like cluster" was inappropriate and would make a confusion. How were the extracellular matrix genes expressed in these clusters? This reviewer was wondering if C3 might represent some sort of intermediate cell-type between CMs and cardiac fibroblasts (Dev Cell. 2016, 39:480). In any rate, more detailed explanations for this cluster are required.

7) BMP4 was used here as an epicardial progenitor marker (Fig. 2B and Supple. Table 1), however, it is known that BMP4 is also closely associated with cardiac progenitor (especially, the 2nd heart field-derived) population (Science. 2015, 348:aaa6071). This is supported by the findings of the recent scRNA-seq paper studying human cardiac development (Dev Cell. 2019, 48:475). Thus, this reviewer is curious about how the expression pattern of BMP4 in the myocardial lineage including a myocardial progenitor population was (Fig. 2H)?

8) ACTA2 was used here as a vascular smooth muscle cell (VSMC) marker (Fig. 2B and Fig. 4A), however, this marker is not specific to the VSMCs but expressed in other cell-types (fibroblasts, etc.). If the authors want to define the VSMC population, the Feature plots for other markers more specific to VSMCs (e.g., MYH11, TAGLN, PDGFRB, etc.) should be involved.

9) The TMSB4X cluster was divided into the three sub-clusters by UMAP projection (Fig. 4A), and the authors defined the cluster 1, 2, and 3 as epi-/endo-/myo-cardial, vascular smooth muscle-like, and vascular endothelial-like cells, respectively. However, as some "lineage-specific" markers (e.g., NPC2) were overlapped among clusters, this definition is somewhat obscure. The resolution of the Feature plots (Fig. 4A, right) is also low, and it is difficult to see which cells in each cluster were (co-)expressing those "lineage-specific" markers.

10) The authors identified significant upregulation of not only cytoskeleton-associated and calcium binding genes but also FKBP1A and PPIA in the TMSB4X cluster, and concluded that the TMSB4X enriched cells play an important role in ventricular development by exhibiting a migratory and proliferative phenotype. However, there was no following mechanistic experiments to validate some molecular and functional connection between such identified TMSB4X-specific genes and thymosin beta-4.

11) The resolution of immunostaining photos (Fig. 4E and Supple. Fig. 5E) is quite low. The high-resolution photos with higher magnification should be included there.

List of experimental data included in the revised manuscript.

- **Fig. 2E-F** displays single molecule FISH (smFISH) images characterizing the epicardial progenitor 2 cell type with mesenchymal phenotype on day 7 as requested by all reviewers. An image panel with nine new image panels was added to **Fig. 2E-F** validating the TCF21+ POSTN+ epicardial origin cells to be LUM- but AGRN+ supporting their mesenchymal phenotype.
- Reviewer #2 suggested confirming that the TCF21+ POSTN+ cells are of epicardial origin as opposed to endocardial origin. We performed smFISH on day 7 hearts to validate the TCF21+ POSTN+ cells to NPR3- (a ventricular endocardial marker) and TEK- (an endocardium derived TCF21+ fibroblast marker). These results are included as **Sup. Fig. 4**.
- Reviewer #2 suggested clarifying if the dendritic cells and macrophages captured in scRNAseq could be circulating or tissue resident or both. We have added three new panels in **Sup. Fig. 2** with cell type prediction scores for immune cell types and erythrocytes suggesting dendritic cells to be circulating and presence of tissue resident macrophages in the spatial RNA-seq.
- All reviewers requested firm validation experiments showing TBX5 and TBX20 expression differences between left and right ventricles. We performed smFISH on heart tissue from day 7 and day 14 and stained for TNNC1, TBX5, and TBX20 to compare the expression between RV and LV. We have included the FISH images in **Fig. 3I**, in support of our original claim.
- Reviewer 1 suggested that the proportion of endothelial cells decrease due to compaction of the ventricular wall and due to the thickening of the compact myocardium. We performed fish on day 7 and day 10 hearts to show these processes, in agreement with the reviewers' suggestions. FISH images included in **Sup. Fig. 6E**.
- All reviewers requested FISH-based validation of TMSB4X high cells characterization across stages. We stained TNNC1, CDH5, and TMSB4X confirming the upregulation of TMSB4X in vascular endothelial cells on day 7, day 10, and day 14. These images have been included in **Fig. 5C-F**. We also stained PDGFRB, CDH5, and TMSB4X showing upregulation of TMSB4X in vascular mural cells and included these images in **Sup. Fig. 8**.
- All reviewers requested high quality immunostaining images to confirm the TMSB4X results. We stained MF20 with CDH5, ACTA2, and TMSB4X on sister sections to characterize the TMSB4X high cell population. These images have been included as **Sup. Fig. 9**.
- Reviewer #3 suggested to look at the expression of congenital heart defect (CHD) related genes and present their relationships with cell type and spatial location within tissue. We have included spatial expression plots for anatomically restricted CHD associated genes in **Fig. 4**.

- Additional feature plots for various lineage specific genes and dot plots for top 10 differentially expressed genes have been added as **Sup. Fig. 3A & B** for epicardial lineage, **Sup. Fig. 5D** for endocardial lineage and **Sup. Fig. 5I** for myocardial lineage as requested by Reviewers. Dot plot for top 5 differentially expressed gene for all cell types in scRNAseq data has been added as **Sup. Fig. 1G**.

Point-by-point address of the specific comments raised by the reviewers.
(Our response in blue font)

Reviewer #1 (Remarks to the Author):

This is a very interesting paper that uses the chick model of heart development to analyze spatiotemporal gene expression in the heart over developmental stages. Developmental time points chosen range from tubular heart stages to the fully formed heart. The four time points chosen are adequate. In order to get both spatial information and single cell data, the authors use a combination of spatial RNAseq and single cell RNAseq (scRNAseq). Spatial RNAseq localizes expression to spots, but loses single cell resolution; scRNAseq has single cell resolution, but loses localization. From a theoretical perspective the project makes sense. Spatial spots are analyzed with RNAseq, and, given the gene expression and location it should be possible to infer the cell composition of the spot, and from there, one can use information from scRNAseq to complete mappings of expression and cell interactions. This is a very exciting project. However, while the idea is sound, it has not been validated. Why not using FISH or a similar technique to validate some of the findings? With FISH it is possible to get single cell resolution of gene expression and thus predicted clusters of cells, including their location and maturation, could be verified. This validation will really strengthen the paper.

Also, it would be nice to verify some findings against previous work in the literature (avian or mouse). Many of the developmental/gene expression processes (including cell differentiation and migration) that take place during heart development are already known and well studied. Do the findings fit well into known facts of heart development? This comparison and discussion will further strengthen the paper.

Thank you for the appreciation of our work and the valuable comments and suggestions that have allowed us to significantly strengthen our paper. We have performed extensive validation experiments, based on multiplexed single molecule FISH (smFISH) using a state-of-the-art Hybridisation Chain Reaction (HCR) v3 technology (H. Choi et al Development, 2018) to validate all our findings. We provide an overview of these validation experiments and new analyses below. In addition, we have improved the discussion of our work in the context of prior literature.

Some minor comments:

1. I had a difficult time understanding exactly how the spatial RNAseq system works (e.g. how big are the spots? How many cells is it sampling? how is the 'material' collected/processed? Why does the preparation involve H&E staining? are there issues with processing, and how were they minimized?).

We now provide additional detail about the spatial RNAseq system (10x Genomics Spatial RNAseq Visium platform) used in the methods section of the revised manuscript. A 10x Genomics Visium Gene Expression slide has 4 capture areas each with an array of 5000 circular spots with printed DNA oligos for mRNA capture. These oligos on each spot have a PCR handle, unique spatial barcode, Unique Molecular Identifier (UMI), and a poly-dT-VN tail for capturing the 3' end of mRNA molecules. Each spot with a unique spatial barcode is 55 μm in diameter and the center to center distance between the spots is 110 μm . One 55 μm spot captures mRNA from 10-20 cells depending on cell size and packing density which is variable across the tissue. We embedded fresh hearts from four stages in OCT, which were then snap-frozen using an isopentane liquid nitrogen bath. Tissue sections from fresh frozen chicken embryonic hearts were mounted for 4 stages with one stage per capture area. These sections were then fixed using pre-chilled methanol for 30 minutes and then H&E stained and imaged. H&E staining was later used by the 10x Genomics Cell Ranger software to detect spots covered by tissue. The 10X Visium technology is robust, we did not encounter any major issues with sample handling and data generation.

2. For the scRNAseq collection, several hearts are pooled together. How exactly was this done to minimize changes in expression in cells? Typically the tissue has to be collected and analyzed right away not to lose expression. But collecting a dozen or more hearts takes time. I did not see a discussion on this or possible effects of collection methods.

After a lot of practice with isolating hearts from early stages, we were able to isolate several dozen hearts in around an hour. The isolated hearts were placed in Hank's Balanced Salt Solution on ice and then digested in 1.5 mg/mL collagenase type II/dispase (Roche) for one cycle of 20 minutes and one cycle of 10 minutes under mild agitation at 37°C. Cells from each sample were stained with Trypan Blue and the cell viability was determined before loading the cells on the 10x Chromium. All samples had a high cell viability (87 to 95%). We performed data preprocessing to remove potential dead cells (cells with less than 200 genes detected and more than 30% mitochondrial transcripts removed, as is standard in the field). We now provide additional detail on the experiments in the methods section of the revised manuscript.

3. There are no 'fetal' stages in avian - those would be embryonic stages instead. However, you can say it is a fetal-like stage.

We agree with the reviewer and have changed the terminology in the manuscript accordingly.

4. Lines 154-156. When I first read it, it seemed to me that 'epicardial lineage', which is all over the heart, meant cells of epicardial origin (in general), which abound in the heart. So the data made sense. However, later on it is stated that epicardial lineage is with respect to the tubular heart stage (HH24) - so cells derived from epicardial cells at HH24. Which one is right? I don't think that you will find epicardial cells (derived from the HH24 epicardium) in the septum of the heart later on... But of course you will find neural-crest cell derived cells, which are of epicardial origin. Please clarify.

Thank you for the comment. In the new version of the manuscript we clarified that the epicardial lineage group discussed represents TCF21+ POSTN+ TEK- NPR3- cells in the ventricular free wall.

5. Could you verify the statement on lines 154-156 with previous literature findings?

Previous literature has shown that ventricular endocardial cells can differentiate into a small percentage of fibroblasts and mural cells within the ventricular free walls and that sinus venosus-derived endocardial cells make up the majority of coronary vascular endothelial cells within the ventricular free walls (Zhang et al *Circulation Research*, 2018). Nonetheless, transitioning phenotypes are not represented in our dataset, likely because these cells are very rare and short-lived (**Sup. Fig. 5A**). For the myocardial lineage, it is well known that differentiation occurs very early in development (Cui et al *Cell Reports*, 2019), and this is reflected in the differentiation trajectory we observed for the myocardial cell lineage (**Sup. Fig. 5F & 5G**). We observed rich spatiotemporal differentiation in the epicardial lineage that was further supported by multiplexed smFISH validation experiments. We have updated our manuscript with the above references.

6. I am not sure I understand what the pseudo-time means in the context of the paper. Could you please clarify?

Individual cells represented in scRNAseq data are executing gene expression programs in an unsynchronized manner. We used two bioinformatic trajectory reconstruction tools, PHATE and monocle, which learn the structure in single cell gene expression data and use the changes in gene expression to order single cells based on "pseudotime" (term introduced in: Trapnell et al *Nature Biotechnology* 2014), placing them along a trajectory corresponding to a biological process such as cell differentiation. We clarify this point in the revised manuscript.

7. Lines 190 to 196. Could you independently validate this finding? It should be relatively straightforward to mark the epithelial cells at HH24 and then follow them to see if they behave as described. Perhaps another group has already done similar work on this too.

Thank you for this comment. The HH24 (day 4) heart stage contains epicardial progenitor cells expressing several epicardial progenitor markers. We observe significant differentiation heterogeneity and a bifurcation in the epicardial lineage at day 7, which we further validated with multiplexed smFISH assays. We used smFISH to visualize gene expression in day 7 tissues (**Fig. 2E-F and Sup. Fig. 4**). In agreement with the epicardial progenitor -2 cluster observed in our scRNA-seq data, smFISH results show that there are TCF21+ POSTN+ NPR3- and TCF21+ POSTN+ TEK- cells present within the compact myocardium, supporting our observation that there exists an epicardial progenitor-derived cell (EPDC) population at day 7 (HH31) that is undergoing epithelial-to-mesenchymal transition (SNAI1+, according to our scRNAseq data) (**Sup. Fig. 3B**). In addition, we stained for LUM (outermost epicardial layer marker) and demonstrated that these epicardial progenitor-2 cells are also LUM- but there exists a large epicardial progenitor-1 cell population that is TCF21+ LUM+ and lines the outer epicardial layer of the heart. Lastly, we validated that these epicardial progenitor-2 cells are AGRN+, a key extracellular matrix protein implicated in migration. Just recently, a biorxiv study from Tzahor's group demonstrated agrin is critical for epicardial EMT processes (Sun et al BioRxiv, 2020) and our results corroborate these novel findings. Previous studies by Lupu et al 2020 and Xiao et al 2018 have revealed an epicardial transitioning phenotype during key stages of epicardial derived cell EMT and differentiation in mice, which our results also support. These more detailed observations along with additional validation experiments have been included in the revised manuscript (**Fig. 2E-F and Sup. Fig. 4**).

8. Lines 253-255 - proportion of endothelial cells decrease. Also due to thickening of the compact myocardium? while the endocardium continues to be a monolayer.

Thank you for the comment. The average proportion of endothelial cells decreased in both left and right ventricles, likely due to i) a reduction in fenestrated trabeculated myocardium layers with developmental time, and ii) thickening of compact myocardium, where the myocytes and fibroblast cells occupy the bulk of the compact myocardium while the endocardium continues to be a monolayer. We performed multiplexed smFISH on sections from day 7 & day 10 using TNNC1 as cardiomyocytes marker, and CDH5 as endocardial marker to validate this observation. smFISH clearly validates an increase in proportion of myocardial populations and a decrease in the proportion of endocardial population per unit area with developmental time in the ventricular wall due to the reasons listed above. We have included the smFISH images in the revised manuscript (**Sup. Fig. 6E**).

9. Lines 286-287 Natriuretic peptide A (NPPB)? Is this correct?

The chicken natriuretic peptide precursor gene cluster's organization is highly conserved compared with the mammalian NPPB-NPPA cluster. However, the NPPA gene has been lost from the natriuretic peptide precursor gene cluster in chicken, therefore we have manually made the

correction in the Ensembl genome annotation and used NPPB as the gene symbol for Natriuretic Peptide B in chicken.

10. The coronary vasculature is of epicardial origin. Does your data fully reflect this fact? I have not seen a discussion on this, although the data seems to make sense from this point of view.

Thank you for the comment. Previous studies have shown that within coronary vasculature in ventricular free walls, only vascular mural cells are derived from the epicardium. The mural cells captured in our datasets were therefore included in our epicardial lineage analysis. However, vascular endothelial cells are mostly derived from sinus venosus endocardial cells and only a small percentage from ventricular endocardium (Zhang et al Circulation Research, 2018). We did not capture any ventricular endocardium derived cells transitioning to vascular endothelial cells in our dataset, possibly because these cells are rare and short-lived.

11. There is some validation performed for the TMSB4X finding. Why immunohistochemistry rather than FISH?

We have now performed both immunostaining and multiplexed smFISH to validate the TMSB4X findings.

1. Using Immunostaining, we labeled heart sections from all four stages discussed in the paper with Thymosin beta-4 (TMSB4X), sarcomeric myosin heavy chain (MF20) as cardiomyocyte marker, VE cadherin (CDH5) as endothelial cell marker, and alpha smooth muscle actin (ACTA2) as vascular smooth muscle marker.
2. Using multi-color, multiplexed smFISH, we simultaneously visualized expression of Thymosin beta-4 (TMSB4X), Cardiac troponin (TNNC1) as cardiomyocyte marker, VE cadherin (CDH5) as endothelial and vascular endothelial marker and Platelet derived growth factor beta (PDGFRB) as vascular smooth muscle markers, in heart sections from day 4 to day 14.

We have updated the manuscript with our additional data and updated findings to specifically better characterize the “TMSB4X high cells” (**Fig. 5C-F and Sup. Fig. 7F, 8, 9**). In summary, through the corroboration of both our repeated immunostaining and new smFISH results, we find that these TMSB4X high cells in day 4 and day 7 are mainly cells in the epicardium and endocardial origin cells and in day 7, day 10, and day 14 coronary vascular cells but NOT cardiomyocytes at the earlier stages (day 4 and day 7) as previously suggested.

12. Lines 438-444: You may be reading too much into these data... it may just be needed for migration of cells to form different parts...

Thank you for this comment. We have updated the discussion to remove any claims about TMSB4X's mechanistic role in ventricular maturation. We have updated the results section of our manuscript to include our updated findings showing persistent upregulation of TMSB4X during coronary vascular and performed additional IHC and smFISH experiments to characterize and validate the "TMSB4X high" cell population across all stages (**Fig. 5C-F and Sup. Fig. 7F, 8, 9**).

13. Lines 463-465: Ratios of cells cannot be derived from scRNAseq as the 'capture' efficiency of cells depends on cell type...

Thank you for the comment. This is exactly why we haven't calculated changes in ratios of cells from scRNA-seq data alone. Rather, we have inferred changes in ratios of cell types from spatial RNA-seq data, which does not require tissue dissociation and cell capture. Spatial RNA-seq does not provide single-cell resolution, but we show in the paper that this can be mitigated by deconvolution of the local spot transcriptomes with matched scRNA-seq data: the spatial spot expression profile can be considered a weighted mix of the single-cell expression profiles of the cells on the spot. scRNA-seq data for matched tissue can be used to estimate cell type composition for the spots, thereby effectively increasing the resolution of spatial RNA-seq. The resulting cell type prediction scores allow us to compare and analyse relative composition of cell types per spot (per unit area) within and between tissue (**Fig. 3B**).

14. Lines 469-484: It seems that very similar studies have already been performed with human tissues. So what exactly is the novelty of your studies - in terms of technique - should be addressed, and the methods validated.

Thank you for the comment. We discussed the limitations of the previous study on human tissues in the original manuscript. Briefly, the previous study had limitations with respect to detecting rare cell types and identifying spatiotemporal cell-cell interactions due to the limited cell types being detected (total of only 3,717 cells), limited genes being used in the In-Situ Sequencing (ISS) panel, and low resolution of the spatial transcriptomic technique (3,115 spots containing ~30 cells per spot). In terms of additional technical novelty, we introduce computational techniques in this paper to analyze and integrate scRNA-seq and spatial RNA-seq datasets to estimate local cell type compositions and visualize developmental time on spatial maps, which has not been done before. This allowed us to visualize differentiation trajectories in a morphological context. In the new manuscript we furthermore report extensive validation of the methods used, based on smFISH and immunostaining.

--

Reviewer #2 (Remarks to the Author):

The authors use scRNAseq and spatial transcriptomic technologies to evaluate changes in cellular composition and spatial restriction of gene expression during cardiac morphogenesis in the chick heart. From spatially resolved maps of gene expression, they infer interactions between different cell types and differentiation trajectories across developmental stages. Despite generating a valuable resource, the authors provide very little experimental validation to support their claims and the study offers limited novel insight into biological processes or regulatory mechanisms. Interpretation of the data is heavily dependent on the sequencing quality, integration and trajectory tools used in their analysis pipeline. Thus, the predictions generated by these approaches must be rigorously tested experimentally. As well as confirming putative expression patterns, ideally at the protein level, genetic lineage tracing and perturbation experiments are required to corroborate and challenge the proposed trajectories.

Thank you for the appreciation of our work and the valuable comments and suggestions that have allowed us to significantly strengthen our paper. We have performed extensive validation experiments, based on multiplexed single molecule FISH using a state-of-the-art Hybridization Chain Reaction (HCR) v3 technology (H. Choi et al Development, 2018). We provide an overview of these validation experiments and new analyses below.

Major concerns

- Spatially resolved gene expression requires validation on histological sections, of mRNA or ideally protein, co-localisation with cell-type specific markers to support the main findings e.g. expression distribution between left vs right ventricle; compact vs trabeculated myocardium. In particular, where data are potentially novel (e.g. C3 cardiomyocyte precursor-like cells) and where data appear to contradict previous findings (e.g. TBX20), experimental validation by multiple approaches is particularly important.

Thank you for the suggestion. To address this comment, we performed multiplexed smFISH on heart sections from day 7 to day 14 (T-box transcription factor 5 (TBX5), T-box transcription factor 20 (TBX20), and cardiac troponin (TNNC1)) and validated the spatially resolved gene expression of TBX5 and TBX20 observed in spatial RNA-seq. We have included the smFISH images in the main manuscript (**Fig. 3I**). The cardiomyocyte precursor-like cells (C3 subcluster in myocardial lineage analysis) captured mainly from day 4 and day 7 hearts expressing maturity markers like TNNC1 were only upregulating hemoglobin transcripts and had a low overall transcriptional activity (**Sup. Fig. 5F & 5J**). We have updated our analysis of cardiomyocyte precursor-like cells as they appear to be transcriptionally less active but differentiated cardiomyocytes from day 4 (HH24) and day 7 (HH31) staged hearts in our dataset.

- Inferred trajectories denote transcriptional similarity, not necessarily bona fide biological processes. The authors should, therefore, validate the conclusions of their trajectory analysis, using immunohistochemistry/in situ hybridization, lineage tracing and genetic perturbation experiments. That said, in support of their conclusions, most of the trajectories described have been previously reported and are unfortunately not novel.

Thank you for these suggestions. In response to this comment, we performed extensive multiplexed smFISH experiments to validate both the lineage analysis and inferred trajectories and we have included the data and analysis in the revised manuscript (**Fig. 2E-F and Sup. Fig. 4**). The fact that many of the trajectories we reported are in agreement with previous literature provides confidence in the methodologies proposed and used in this work. We have not followed the suggestion to perform lineage tracing experiments for three reasons: First, lineage tracing is not possible in the chick model using conventional lineage tracing Cre-drivers. Lineage tracing in chick has been achieved by only a few groups with chick quail chimera technology or CRISPR systems, both of which require manipulating individual embryos, which can introduce variability. Second, lineage tracing is not a perfect method and can lead to labeling of cells that do not actually belong to that specific lineage. Third, with multiplexed smFISH, we were able to simultaneously visualize expression of established lineage markers and the expression of trajectory-specific genes identified by scRNAseq.

- How were epicardial progenitor cells defined? Tcf21 alone is insufficient to identify epicardium and epicardium-derived cells in the absence of lineage tracing tools. Epicardial-derived mural cells do not express Tcf21 and, moreover, non-epicardial derivatives e.g. endocardium-derived fibroblasts also express Tcf21. Thus, trajectory analysis will be confounded by omission of cells that have lost the precursor marker upon differentiation.

To define epicardial progenitor cells, we used multiple epicardial markers TCF21, WT1, TBX18, and ALDH1A2. We found that these markers are almost exclusively expressed in the epicardial progenitor cells-1 and epicardial progenitor cells-2 clusters (**Fig. 1D, Fig 2B, and Sup. Fig. 3B**). In addition, we found that the epicardial progenitor-2 cells (which were not present in the outermost layer of the ventricular free walls but instead were present in various locations in the myocardium at day 7) were NPR3-, NPC2-, and TEK- confirming that these were not of endocardial origin (**Fig. 1D and Sup. Fig. 2**). We performed extensive multiplexed smFISH experiments on day 7 (HH31) heart sections and show that there are TCF21+ POSTN+ epicardial progenitor- 2 cells within the compact myocardium that are NPR3- (well-known ventricular endocardium marker) and TEK- (endocardial derived TCF21+ fibroblast cell marker, Moore-Morris et al J Clin. Invest. 2014) further confirming that these cells are not of endocardial origin. We have included these images in the revised manuscript (**Sup. Fig. 4**).

On a related note, we also demonstrate in the new manuscript that unlike epicardial progenitor-1 cells, epicardial progenitor-2 cells are LUM- (outermost epicardial layer marker in early heart developmental stages) and AGRN+ (a key extracellular matrix proteins implicated in epicardial EMT migration, Sun et al BioRxiv 2020), corroborating our scRNAseq analysis results. Previous studies by Lupu et al 2020 and Xiao et al 2018 have revealed an epicardial transitioning phenotype during key stages of epicardial progenitor derived cell EMT and differentiation in mice, which our results also further support. These more thorough descriptions of the epicardial cells have been included in the manuscript along with additional results/figures to demonstrate/support our findings (**Fig. 2E-F and Sup. Fig. 4**).

- As an example of incorrect conclusion based on inappropriate methodology, the authors conclude that the endocardium undergoes a maturation process, rather than differentiating into other cell types. This is at odds with lineage tracing studies which demonstrate that the endocardium can adopt multiple fates (endothelial, cardiac/valve fibroblasts and mural cells) but can be explained by the fact that authors used only the endocardial cluster in their trajectory analysis (Supp Fig. 1F).

Thank you for this comment. We incorrectly used the term maturation. We agree with the reviewer that there are fate specification processes in endocardium lineage. What we meant to say is that the magnitude of the changes in the transcriptional programs of endocardial cells is small compared to cells from the epicardial lineage (**Sup. Fig. 5A-E**). We have moved the trajectory figure for endocardial cells to the supplement and have updated the discussion of this trajectory analysis to clarify this in the revised manuscript. We agree that cardiac endocardium can adopt multiple fates including endothelial cells, cardiac fibroblast, and mural cells.

- I don't understand why the authors excluded vascular endothelial cells from trajectory analysis "since their lineage origin remains debated". Firstly, there is now a reasonable consensus view that these cells derive from sinus venosus and endocardium (in chick and mouse). Secondly, if the authors are confident in the power of their data to resolve differentiation trajectories, they have an ideal opportunity to evaluate possible contributions from other lineages (e.g. epicardium).

Thank you for this comment. As you correctly pointed out, the coronary vascular endothelial cells are mostly derived from sinus venosus and ventricular endocardium while the mural cells as well as fibroblasts are mostly derived from the epicardium in the ventricular free walls (Zhang et al Circulation Research 2018). However, we were unable to make any specific new conclusions about the origin of vascular endothelial cells and their corresponding differentiation trajectories as these cells were not expressing origin specific gene markers like NPR3, NPC2, TEK, NFATC1 in our scRNAseq dataset (**Fig. 1D**). We did not capture any of the transitioning cells that are differentiating from their endocardial origin, likely because they are rare or short-lived. We clarify this point in the revised manuscript.

- In the absence of experimental validation, statements such as “and were thereby able to reveal how cellular differentiation and morphological changes co-occur” and “suggests its potential morphogenetic role in directing epicardial and endocardial-derived cell migration into the myocardium to initiate and maintain ventricular compaction and maturation.” and (in relation to thymosin beta-4) “our analysis clarified its role in ventricular compaction and maturation” are entirely unjustified. Mechanistic experiments are needed to support these claims.

Thank you for these comments. In the new version of the manuscript, we removed any mechanistic claims and suggestions that were not supported by mechanistic experiments from the discussion.

- A number of relevant studies have not been cited and discussed. As examples, left vs right ventricle and compact versus trabecular myocardial markers in relation to e.g. Li et al 2019, Yvanka de Soysa et al 2019, Li et al 2016; epicardial trajectory model should be discussed in light of Xiao et al 2018 and Lupu et al (2020).

Thank you for these suggestions. We have included these references in the updated version of the manuscript.

Minor concerns

- Fig1B: There are fewer CMs/endocardial cells at D14 samples, is this down to low cell and sequencing depth?

Thank you for this comment. We analysed a total of 5653, 8463, 5190, and 3009 single cell transcriptomes from day 4, day 7, day 10, and day 14 respectively. D14 samples had fewer cells from all cell types compared to other samples. The reduced fraction of myocytes and endothelial cells in the D14 sample is further due to the increase in fraction of fibroblasts, vasculature cells, and mural cells leading to an increased overall cell type heterogeneity in D14 hearts (as shown in the figure below). However, it is not a good practice to compare fractions/ ratios of cells from scRNAseq data as different cell types have different capture efficiency. This is exactly why we haven't calculated changes in ratios of cells from scRNA-seq data alone in our study. Rather, we have inferred changes in ratios of cell types from spatial RNA-seq data, which does not require tissue dissociation and cell capture. Spatial RNA-seq does not provide single-cell resolution, but we show in the paper that this can be mitigated by deconvolution of the local spot transcriptomes with matched scRNA-seq data: the spatial spot expression profile can be considered a weighted mix of the single-cell expression profiles of the cells on the spot (**Fig. 1C and Sup. Fig. 2**). scRNA-seq data for matched tissue can be used to estimate cell type composition for the spots, thereby effectively increasing the resolution of spatial RNA-seq (Explained in Methods). The resulting cell type prediction scores allow us to compare and analyse relative composition of cell types per spot (per unit area) within and between tissue (**Fig. 3B**).

- Fig S1A: Almost 10-fold fewer transcripts per cell at d14, compared with d7, which may result in sequencing saturation. Moreover >2000 genes/cell is required for reliable GO analysis and inferences re: biological processes.

Thank you for your comments. scRNA-seq on the left and right ventricular samples from day 10 and day 14 was performed using 10x Chromium v2 chemistry. We agree with the reviewer that we observe significant differences in sequencing saturation among samples due to differences in chemistry, number of cells captured, and number of samples sequenced per lane on flow cell. We observe a sequencing saturation of ~31% for day 4, ~50% for day 7 ventricles, ~91% for day 10 ventricles, and 94% and ~78% for day 14 ventricles. Gene ontology analysis was performed using the differential gene expression analysis results and we used stringent p-value cutoff (p-value < 10^{-15}) to select genes for enrichment analysis.

- Total of 22,315 cells sequenced. The authors need to detail exact cell numbers per stage in the methods.

We analysed a total of 5653, 8463, 5190, and 3009 single cell transcriptomes from day 4, day 7, day 10, and day 14 respectively. We now include these details in the methods section.

- After scanorama batch correction and integration, is there evidence of over-correction?

Thank you for this comment. Scanorama has been shown to be robust to different dataset sizes and sources, preserves dataset-specific populations, does not require that all datasets share at least one cell population, and preserves biological differences in time course experiments (Hie et al Nature Biotechnology, 2019). To verify this, we compare and visualize cell type clusters on UMAP dimensions before and after integration. Our analysis indicated that cell clusters representing all three lineages had minimal overlap before integration suggesting significant biological differences between early and late stages (**Sup. Fig. 1C, left**). Scanorama based integration corrected for batch differences particularly in myocardial cells (**Sup. Fig. 1C, right**). However, unsupervised clustering of scRNAseq dataset revealed distinct clusters for NKX-2.5+ PITX2+ IRX4+ immature myocytes cluster mainly from day 4 and day 7, and NKX-2.5- TNNC1+ MYL10+ mature cardiomyocyte cluster mainly from day 10 and day 14, suggesting that biological differences were preserved (**Fig. 1B, 1D**). In endocardial and epicardial lineages, there was minimal overlap between early and late stages even after scanorama integration (**Sup. Fig. 1C, right**), suggesting that biological differences were preserved and that scanorama did not overcorrect while removing batch effects.

- The authors should present heatmaps with the top 5 or top 10 differentially expressed genes for each cluster, to allow confirmation (by readers) on appropriateness of cluster annotation. The

markers highlighted are not necessarily the better characterised/ more selective. Why not e.g. Npr3/Nfatc1 for endocardium; Wt1/Upk3b for epicardium?

In the new version of the paper, we include Dot Plots showing top 5 differentially expressed genes (**Sup. Fig. 1G**). The expression of these genes along with some known canonical markers (**Fig. 1D**) was used to assign cell type labels to the clusters. NPR3 and NFATC1 were mainly expressed in endocardial cells, while WT1 was expressed in Epicardial progenitor-1 cells. Upk3b gene expression was not detected in our dataset.

- Fig2B and sup. Fig3A: the authors should discuss the differences between PHATE and Monocle 2 trajectory inferences for epicardial progenitor cells.

Monocle 2 uses a technique called reversed graph embedding to learn the structure of the manifold that describes a single-cell experiment. It employs DDRTree: discriminative dimensionality reduction via learning a tree to implement the general framework of reversed graph embedding. On the contrary, PHATE captures both local and global nonlinear structure using an information-geometric distance between data points and produces lower-dimensional embeddings that are quantitatively better denoised as compared to existing visualization methods. In the epicardial lineage, PHATE visualization revealed progenitor heterogeneity and a branching cell fate at day 7 which later merged into a differentiated fibroblast cell type. Although monocle 2 was able to capture the day 7 heterogeneity on distinct branches it was unable to conserve the global loop structure with day 4 branching into two distinct day 7 populations and merging again into mature populations from day 10 and day 14.

- Figure 3B: Is cell-type composition based on the sections shown? Multiple independent samples and sections from different regions are required for this to be accurate and validation is required. The D14 data derive from one section, which cannot be representative of the whole heart, given the dorsal-ventral variation in cellular composition. The differing numbers of sections per stage will affect comparison between stages. This limitation needs to be acknowledged.

The cell type compositions were estimated by averaging the cell type prediction scores for spots in different anatomical regions in sections shown. We had multiple sections from biological replicate hearts for day 4 (n = 5), day 7 (n = 4), and day 10 (n = 2) but only one section for day 14 (n = 1). In our Spatial Gene Expression experiment (explained in Methods), we used one capture area containing 5000 spots for each developmental stage and accommodated multiple (classic four-chamber view) sections on the capture area based on the size of the tissue at that stage. Overall, tissue sections from day 4 to day 14 covered a total of 747, 1966, 1916, and 1967 spots on the capture area, respectively. We agree that multiple independent samples and sections from different regions are required for this to be representative of the entire heart. To validate the cell type composition findings, we performed multiplexed smFISH to visualize expression of

cardiomyocyte marker *TNNC1*, and endothelial marker *CDH5* on sections from three to four independent hearts across developmental stages (**Sup. Fig. 6E**). These experiments validated our initial observation of an overall increase in the proportion of myocytes and a decrease in endocardial cells per unit area in the ventricular wall with developmental time, in agreement with our cell type composition results. Nonetheless, we agree with the reviewer that these numbers reflect specific sections and not the entire organ and we have acknowledged this limitation in the methods sections of the revised manuscript.

- If the TMSB4X high cell cluster comprises a heterogeneous population of cells, it is difficult to comprehend why they cluster together, rather than with the constituent cell populations. Do they still cluster if embryonic stages are analyzed separately? The authors should demonstrate TMSB4X expression levels across clusters at separate stages. It is unclear whether TMSB4X expression is consistently high across all stages or primarily contributed by increased expression at D14 (FigS5B).

We performed unsupervised clustering of transcriptomes using Louvain algorithm on a k-nearest neighbors graph constructed using gene expression data as implemented in Seurat-v3. Unsupervised clustering revealed a “TMSB4X high cells” cluster. Although this cluster contains multiple different cell types from different developmental time points, they have a distinct transcriptional program. In other words, these TMSB4X cells are more similar to each other than they are to the remaining TMSB4X low cell type clusters. We verified that, if the embryonic stages are clustered separately, the TMSB4X high cells still form a cluster that is distinct from the TMSB4X low population within that stage. Convergence of Louvain clustering algorithm and thus number of clusters obtained also depends on the clustering resolution which can be increased to look at the smallest clusters of cells. To address the last part of the question, we have included a plot showing TMSB4X expression levels across clusters at separate stages in the revised manuscript (**Sup. Fig. 7A**). We also performed extensive smFISH and immunostaining experiments to visualize gene expression in TMSB4X high cells across stages and have included representative images in the revised manuscript (**Fig. 5C-F and Sup. Fig. 7F, 8, 9**).

- Macrophages and dendritic cells are described as “circulating”. Can the authors exclude that some may be cardiac-resident?

Thank you for pointing this out. We could not exclude the possibility that some of these immune cells are cardiac resident as opposed to circulating and therefore we have dropped the term circulating.

- “Smooth muscle cell” clusters should be described as mural cells as the majority at these stages will be pericytes and immature smooth muscle progenitors.

Thank you for this suggestion. We have updated the manuscript accordingly.

- Nppb should be defined as natriuretic peptide B, not A (presumably, in trabeculae, as opposed to Nppa, which is natriuretic peptide B).

The chicken natriuretic peptide precursor gene cluster's organization is highly conserved compared with the mammalian NPPB-NPPA cluster. However, the NPPA gene has been lost from the natriuretic peptide precursor gene cluster in chicken, therefore we have manually made the correction in the Ensembl genome annotation and used NPPB as the gene symbol for Natriuretic Peptide B in chicken.

- Figure 3B: The key does not indicate the cell type label represented by grey bars.

Thank you for pointing this out. We have fixed this error.

- Figure 4A: Typo Cluster 1 should be Cluster

Thank you for pointing this out. We have fixed this typo.

--

Reviewer #3 (Remarks to the Author):

The authors have reported a detailed atlas of the diverse cellular lineages in the developing chicken heart and revealed their spatial organization and interactions during cardiac development by using new approaches to combine high-throughput single-cell RNA-sequencing (scRNA-seq) with spatially resolved RNA-seq. Through novel anchor-based bioinformatic approaches, they delineated molecular and morphological aspects of cardiac development, including dynamic cellular interactions within heart tissue, changes in cellular composition and anatomically restricted gene expression at high spatiotemporal resolution. Finally, they found a stage-dependent role of thymosin beta 4 in the coordination of multi-lineage cellular populations for ventricular development, i.e., compaction and maturation. Although scRNA-seq has been used to date as an excellent tool to identify cellular hierarchies and molecular signatures in developmental biology (and other settings as well), it is a caveat that it does not preserve spatial information about cellular interactions and tissue morphology. Taken together, the approaches to integrate spatial and single-cell transcriptomic data, conducted in this paper, likely hold promise to identify key stage- and anatomical location-specific machinery programs that regulate organ development in embryogenesis, as their analysis appears to provide novel insights into several regulatory programs that guide ventricular development and morphogenesis. Overall, the research and manuscript described here were well designed and described; however, the several critical issues which need to be addressed should be pointed as follows.

Thank you for the appreciation of our work and the valuable comments and suggestions that have allowed us to significantly strengthen our paper. We have performed extensive validation experiments, based on multiplexed single molecule FISH using a state-of-the-art Hybridization Chain Reaction (HCR) v3 technology (H. Choi et al Development, 2018) to validate all our findings. We provide an overview of these validation experiments and new analyses below.

1) The UMAP projection of 22,315 scRNA-seq of fetal chicken hearts at four developmental stages revealed 15 distinct cell type clusters (Fig. 1B), which included epi-, endo-, and myocardial developmental lineages. However, to this reviewer, it is a bit surprising that the 15 clusters did not exhibit clearly another important anatomical differences, i.e., atria and ventricles, unlike the previous similar papers studying scRNA-seq in embryonic cardiogenesis (Dev Cell. 2019, 48:475; Cell. 2019, 179:1647). Could the authors find any differences of gene expression patterns between atria and ventricular cardiomyocytes (CMs) or do sub-clustering of them among the myocardial lineage (Supple. Fig. 1F)?

Thank you for this comment. The scRNAseq datasets were generated using ventricular tissue only (whole ventricle in day 4 hearts and left and right ventricular free walls separately in day 7, day 10, and day 14). Therefore, we did not see any clusters for anatomical regions like atria and ventricles in the myocardial lineage. However, our spatial RNAseq data contains full heart sections with all four chambers for developmental stages day 7 to day 14. As discussed in the paper, unsupervised clustering of spot transcriptomes from spatial RNA-seq revealed separate clusters for atria and ventricular tissue. We also observed spatially restricted gene expression patterns in these anatomical regions (**Fig. 3A and Sup. Fig. 6A**).

2) Another concern in the UMAP projection of 22,315 scRNA-seq (Fig. 1B) is that the 15 clusters included some odd one, i.e., cell population expressing high levels of thymosin beta-4 (TMSB4X), which consisted of 3 sub-clusters including CMs, endo- and epi-cardial cells, and vascular smooth muscle- and endothelial-like cells (Fig. 4A). What's the rationale about grouping the "TMSB4X" cluster together with other definitive 14 cell-types in the first UMAP projection (Fig. 1B)? It might provide readers with a bit of confusion.

Thank you for this comment. We performed unsupervised clustering of 22,315 transcriptomes using the Louvain algorithm on k-nearest neighbor graph as implemented in Seurat-v3 (detailed in Methods). Unsupervised clustering revealed a "TMSB4X high cells" cluster with upregulated Thymosin beta-4 transcript expression (**Fig. 1B and Fig. 5A**). Although this cluster consisted of multiple different cell types from different developmental time points, they have a unique transcriptional program. In other words, these TMSB4X cells are more similar to each other than they are to the remaining TMSB4X low cell type clusters, and as a consequence they cluster together. We also included differential gene expression analysis results for the TMSB4X high cell

cluster revealing genes involved in cytoskeleton organization and calcium signaling (**Fig. 5B**). We have clarified this in our revised manuscript.

3) This type of a resource paper featuring scRNA-seq data analysis is often hampered by its descriptive nature with less informative discoveries that significantly advance our notions and understandings in developmental biology. One way to resolve this weak point is to investigate several disease-caused genes' distribution and expression patterns in the experimental dataset. In this case, it is highly recommended to examine how human congenital heart disease (CHD) genes were expressed in the spatial and single-cell RNA-seq dataset reported here and to delineate some relationship, if any, between some CHD genes and spatiotemporally specific cell-types.

Thank you for the great suggestion. We performed additional analysis to identify spatio-temporal patterns in the expression of congenital heart defects (CHD) associated genes encoding for various regulatory transcription factors, signaling molecules, and structural proteins. Our spatial RNA-seq analysis revealed the expression of multiple functionally diverse CHD associated genes to be restricted or enriched in specific anatomical regions. This insight could provide a new understanding of the manifestation of disease. We include these results in a new main manuscript figure (**Fig. 4**).

4) In Fig. 2E (right), there seemed like no clear differences about the endocardial or endothelial cell markers' expression among the three clusters (C1, C2, and C3) of the endocardial lineage. Is this correct? If the authors could find any stage (e.g., early C1 or late C3)-specific expression of such genes, it should be shown.

Thank you for this comment. We refined our ventricular endocardial analysis and sub-clustered these cells into two clusters as opposed to three to maximize the differences between subclusters. We have included the sub-clustering results and expression plots of canonical markers (**Sup. Fig. 5A**). We have also added a DotPlot showing top-10 differentially expressed genes for the endocardial subclusters (**Sup. Fig. 5D**).

5) How were other endocardium-specific markers, such as NPR3 and NFATC1 (Circ Res. 2016, 118:1880) expressed in the endocardial lineage analyzed here?

Thank you for this suggestion. We have expanded **Fig. 1D** to include these additional canonical markers. NPR3 and NFATC1 remain consistently expressed by the endocardial lineage across the developmental stages. Interestingly, as independently shown by both scRNAseq data and multiplexed smFISH and supported by recent literature (Zhang et al Circulation Research, 2018), NPR3 is also lowly expressed by the epicardial progenitor cells and valve cells (**Fig. 1D and Sup. Fig. 4B**). NFATC1 is also expressed by valve cells in our single cell data (**Fig. 1D**) and supported by the literature (Wu et al Circulation Research, 2011).

6) In Fig. 2H showing the myocardial lineage, the 3rd cluster (C3) was not clearly defined. This cluster was very heterogeneous and seemed to consist of cells from not only day 4 and day 7 but day 10 and day 14. Thus, the terminology, “a cardiomyocyte precursor-like cluster” was inappropriate and would make a confusion. How were the extracellular matrix genes expressed in these clusters? This reviewer was wondering if C3 might represent some sort of intermediate cell-type between CMs and cardiac fibroblasts (Dev Cell. 2016, 39:480). In any rate, more detailed explanations for this cluster are required.

Thank you for this suggestion. After further investigation into this cluster, we conclude that these C3 cluster cells (renamed as cardiomyocyte-2, Fig. 1B, 1D) are most likely simply cardiomyocytes with an overall lower transcriptional activity (Sup. Fig. 5E). We do not see any evidence for these cells to an intermediate between CMs and cardiac fibroblasts. We have updated the manuscript accordingly.

7) BMP4 was used here as an epicardial progenitor marker (Fig. 2B and Supple. Table 1), however, it is known that BMP4 is also closely associated with cardiac progenitor (especially, the 2nd heart field-derived) population (Science. 2015, 348:aaa6071). This is supported by the findings of the recent scRNA-seq paper studying human cardiac development (Dev Cell. 2019, 48:475). Thus, this reviewer is curious about how the expression pattern of BMP4 in the myocardial lineage including a myocardial progenitor population was (Fig. 2H)?

Thank you for this comment. We referred to the cardiomyocyte cluster expressing NKX2.5, PITX2, IRX4 as myocardial progenitor cells. However, these cells really are immature myocardial cells as opposed to 2nd heart field-derived cardiac progenitors as they express cardiomyocyte specific markers like TNNC1 and ACTC1. In the new version of the manuscript, we have renamed this cluster as immature myocardial cells. Myocardial lineage cells including the immature myocardial cells in the scRNAseq dataset did not express BMP4 at the time points being analyzed in our study (shown below).

8) ACTA2 was used here as a vascular smooth muscle cell (VSMC) marker (Fig. 2B and Fig. 4A), however, this marker is not specific to the VSMCs but expressed in other cell-types (fibroblasts, etc.). If the authors want to define the VSMC population, the Feature plots for other markers more specific to VSMCs (e.g., MYH11, TAGLN, PDGFRB, etc.) should be involved.

Thank you for this suggestion. As expected, we found that vascular smooth muscle cells/ mural cells also express markers like MYH11, TAGLN, and RGS5 (**Fig. 1D**). We have included these markers in a new version of **Fig. 1D** and have used PDGFRB to identify mural cells in vascular networks by multiplexed smFISH (**Sup. Fig. 8**).

9) The TMSB4X cluster was divided into the three sub-clusters by UMAP projection (Fig. 4A), and the authors defined the cluster 1, 2, and 3 as epi-/endo-/myo-cardial, vascular smooth muscle-like, and vascular endothelial-like cells, respectively. However, as some “lineage-specific” markers (e.g., NPC2) were overlapped among clusters, this definition is somewhat obscure. The resolution of the Feature plots (Fig. 4A, right) is also low, and it is difficult to see which cells in each cluster were (co-)expressing those “lineage-specific” markers.

Thank you for this suggestion. We performed additional immunostaining and multiplexed smFISH to further validate/characterize the TMSB4X high cell cluster across all developmental stages. In agreement with our single cell data, we found that the cells expressing high levels of TMSB4X are mostly endocardial and epicardial cells from day 4. However, we did not observe co-expression of cardiomyocyte markers with smFISH, as previously suggested, and coronary vascular endothelial cell and vascular mural cells from day 7 onwards (**Fig. 5C-F and Sup. Fig. 7F, 8, 9**). We clarify this point and report a persistent upregulation of thymosin beta-4 expression in coronary vascular cells across all stages.

10) The authors identified significant upregulation of not only cytoskeleton-associated and calcium binding genes but also FKBP1A and PPIA in the TMSB4X cluster, and concluded that the TMSB4X enriched cells play an important role in ventricular development by exhibiting a migratory and proliferative phenotype. However, there were no following mechanistic experiments to validate some molecular and functional connection between such identified TMSB4X-specific genes and thymosin beta-4.

Thank you for this comment. We agree with the reviewer that evidence of co-expression does not mean there is necessarily a direct functional connection and have removed these claims.

11) The resolution of immunostaining photos (Fig. 4E and Supple. Fig. 5E) is quite low. The high-resolution photos with higher magnification should be included there.

We have performed additional multiplexed smFISH and immunostaining validation experiments and have included higher magnification images at higher resolution in the main manuscript as well as in supplementary material.

Reviewers' Comments:

Reviewer #1:

Remarks to the Author:

This is a revised version of a manuscript submitted by the authors in which they use a combination of localized RNAseq and scRNAseq to investigate heart development trajectories. The main criticism - from my point of view at least - was the lack of proper validation of the methodology presented. The authors, however, have now included extensive validation of results using single molecule FISH, and have more extensively discussed their findings in view of previous works. They have also addressed all the other concerns raised by reviewers. I have no further concerns.

Reviewer #2:

Remarks to the Author:

The authors have addressed many of the original comments thoroughly. In particular, experimental validation of gene expression strengthens their conclusions. However, I find there are still inaccurate and unsubstantiated claims regarding lineage origin and some confusing, unsupported statements, as detailed below.

1. Within the past 1-2 years, several single cell studies of the developing heart have been published. Unfortunately, that detracts from the novelty of the present study. The depth of information provided, combined with the spatial transcriptomic profiling and addition of chicken heart to the collection, alongside the published mouse and human studies, mean this is still a valuable resource, worthy of publication. However, the advances made by the other studies need to be acknowledged and the novelty of the present study should not be over-stated.
2. In the results section, the epicardial subclusters, representing different stages of the differentiation trajectory are very well described. However, the sentences in the abstract and introduction do not reflect this, they are incorrect and misleading. "revealed spatiotemporal transcriptional heterogeneity within the epicardial lineage" and "epicardial progenitor cell heterogeneity" is misleading and adds to the already confusing body of literature, where studies have failed to separate epithelial from mesenchymal and differentiated derivatives of the epicardium and erroneously inferred heterogeneity in the starting population. A strength of the Mantri study is that it adds further clarity to the differentiation trajectory, which has been suggested by some to differ between chick and mouse, although this now appears even more likely to be due to differences in methodologies used. The authors should therefore reword to clarify precisely what they have found (characterised the differentiation trajectories, which is consistent with the bifurcation reported for mouse (refs 46, 47)). I would also suggest renaming the whole heart clusters (Epicardial progenitors 1 and 2) to Epi-epithelial and Epi-mesenchymal, respectively, to avoid any misleading suggestion of heterogeneity in the progenitor population.
3. Whilst I am sympathetic to the difficulties of lineage tracing in chick, demonstrating expression at a particular developmental stage cannot be used to conclude lineage origin. Npr3 is down-regulated in endocardial cells when they adopt an endothelial fate (Zhang et al Circ. Res. 2018, and likely the same will occur in those adopting fibroblast or mural cell fate). Similarly, Tek/Tie2 expression does not distinguish lineage origin. Citing the Moore-Morris study and inferring that a similar methodology was used is misleading as Moore-Morris et al used Tie2-Cre- based lineage tracing to conclude endocardial origin of fibroblasts, not Tek expression. In the absence of lineage tracing data, the authors need to explicitly state the limitations of the model used, reword their conclusions to state that their data are e.g. 'consistent with the possibility that the cell populations described may derive from lineage X, as supported by other studies (cite)...' but refrain from unequivocally concluding origin. I would suggest removing Npr3/TEK smFISH data from the manuscript altogether as they offer very little and are misleading. As the authors highlight, Npr3 is also expressed in epicardium, thus even if lineage tracing had been possible, the origin could not be concluded based on this marker alone.
4. It is also misleading to suggest that endocardial-derived fibroblasts, mural or endothelial cells are rare. Unless the chick is profoundly different, this seems unlikely (and would require definitive

proof). Just because they are not captured, that does not make them rare. The likely explanation is that the transition occurs rapidly and the cells don't exist with a transitional (mixed) marker profile at the stages collected. As such, they would not be captured without lineage tracing or collection of additional, intermediate stages (they have only been captured as endocardium or derivatives, not mid-transition). Again, it is better to acknowledge technical limitations, than to draw potentially erroneous conclusions which risks misleading the field.

5. The data on Thymosin b4 are also interpreted and discussed in a way that is misleading. Although they don't explicitly say so, as written, it is implied that there is little or no expression of Tmsb4x in cardiomyocytes. From their images and published literature, it is clear that there is expression in cardiomyocytes (in keeping with the known ubiquitous expression; only exception being erythrocytes). Qualitative ISH/ immunostaining always demonstrates relative expression; longer exposure would saturate vascular/epicardial signals but reveal abundant expression in cardiomyocytes. This is particularly apparent at the protein level (supplemental figure 9). As expected, protein expression in established cardiomyocytes would be less dynamic than mRNA in the actively developing epicardial and coronary lineages. In the discussion, the authors conclude that discrepancies in the literature reporting Tmsb4x mutant phenotypes may be explained by a failure to target the lineages that most abundantly express the gene. This is entirely unfounded. The studies cited targeted Tmsb4x globally and selectively in endothelium and myocardium (albeit with different methodologies) and still reached discrepant conclusions. Thus, global versus conditional targeting is not the explanation and the discussion requires rewording. Moreover, just because a gene is expressed at a lower level in one cell type compared with another does not necessarily mean a phenotype will be less severe if it is deleted. A further minor point, for clarification, the authors say that "one other beta-thymosin was expressed, Tmsb15" – were the others not expressed e.g. Tmsb9, Tmsb10?

6. It would be helpful for those unfamiliar with the computational methods if the authors could briefly summarise the differences between the Monocle and PHATE for trajectory reconstruction.

7. Minor point: it is unclear what the authors mean by "significantly upregulated expression of midkine" in the epicardium. Up-regulation implies a change in expression (over time, with treatment etc.). Enriched would be more appropriate here to describe differential expression between cell types.

Reviewer #3:

Remarks to the Author:

The authors addressed well the issues and concerns the reviewers had raised. Additional experiments and analyses conducted in the revised manuscript have improved the quality of that much. Now no major issues or concerns are pointed.

Only minor points: Please correct typos and misspelling, such as "PDGRFB" on Line 365, "PDGRB+" on Lines 368 and 369, etc.

Point-by-point address of the specific comments raised by the reviewers.

(Our response in blue font)

Reviewer #1 (Remarks to the Author):

This is a revised version of a manuscript submitted by the authors in which they use a combination of localized RNAseq and scRNAseq to investigate heart development trajectories. The main criticism - from my point of view at least - was the lack of proper validation of the methodology presented. The authors, however, have now included extensive validation of results using single-molecule FISH, and have more extensively discussed their findings in view of previous works. They have also addressed all the other concerns raised by reviewers. I have no further concerns.

We thank the reviewer again for the great suggestions and comments, and for the appreciation of our work.

Reviewer #2 (Remarks to the Author):

The authors have addressed many of the original comments thoroughly. In particular, experimental validation of gene expression strengthens their conclusions. However, I find there are still inaccurate and unsubstantiated claims regarding lineage origin and some confusing, unsupported statements, as detailed below.

Thank you for the appreciation of our work and the valuable comments and suggestions that have allowed us to significantly strengthen our paper.

1. Within the past 1-2 years, several single-cell studies of the developing heart have been published. Unfortunately, that detracts from the novelty of the present study. The depth of information provided, combined with the spatial transcriptomic profiling and addition of chicken heart to the collection, alongside the published mouse and human studies, mean this is still a valuable resource, worthy of publication. However, the advances made by the other studies need to be acknowledged and the novelty of the present study should not be over-stated.

Thank you for your appreciation of our work. We have discussed and cited all the relevant human and mouse studies in the discussion.

2. In the results section, the epicardial subclusters, representing different stages of the differentiation trajectory are very well described. However, the sentences in the abstract and introduction do not reflect this, they are incorrect and misleading. “revealed spatiotemporal transcriptional heterogeneity within the epicardial lineage” and “epicardial progenitor cell heterogeneity” is misleading and adds to the already confusing body of literature, where studies have failed to separate epithelial from mesenchymal and differentiated derivatives of the epicardium and erroneously inferred heterogeneity in the starting population. A strength of the

Mantri study is that it adds further clarity to the differentiation trajectory, which has been suggested by some to differ between chick and mouse, although this now appears even more likely to be due to differences in methodologies used. The authors should therefore reword to clarify precisely what they have found (characterised the differentiation trajectories, which is consistent with the bifurcation reported for mouse (refs 46, 47)). I would also suggest renaming the whole heart clusters (Epicardial progenitors 1 and 2) to Epi-epithelial and Epi-mesenchymal, respectively, to avoid any misleading suggestion of heterogeneity in the progenitor population.

Thank you for your helpful suggestions. We agree with the comments and have made these changes in the revised manuscript. We have removed the terms “epicardial lineage heterogeneity” and “epicardial progenitor heterogeneity”. We have clarified in the abstract and introduction that our analysis identifies transcriptional differences between epithelial and mesenchymal populations within epicardial cells. We have renamed the whole heart clusters (Epicardial progenitors 1 and 2) to Epi-epithelial and Epi-mesenchymal.

3. Whilst I am sympathetic to the difficulties of lineage tracing in chick, demonstrating expression at a particular developmental stage cannot be used to conclude lineage origin. Npr3 is down-regulated in endocardial cells when they adopt an endothelial fate (Zhang et al Circ. Res. 2018, and likely the same will occur in those adopting fibroblast or mural cell fate). Similarly, Tek/Tie2 expression does not distinguish lineage origin. Citing the Moore-Morris study and inferring that a similar methodology was used is misleading as Moore-Morris et al used Tie2-Cre- based lineage tracing to conclude endocardial origin of fibroblasts, not Tek expression. In the absence of lineage tracing data, the authors need to explicitly state the limitations of the model used, reword their conclusions to state that their data are e.g. ‘consistent with the possibility that the cell populations described may derive from lineage X, as supported by other studies (cite)...’ but refrain from unequivocally concluding origin. I would suggest removing Npr3/TEK smFISH data from the manuscript altogether as they offer very little and are misleading. As the authors highlight, Npr3 is also expressed in epicardium, thus even if lineage tracing had been possible, the origin could not be concluded based on this marker alone.

Thank you for your suggestions and feedback. We have now explicitly stated that our analysis is consistent with the possibility that the TEK- NPR3- fibroblast cells are derived from epicardial cells. As suggested, we have removed the NPR3 and TEK smFISH data from the manuscript.

4. It is also misleading to suggest that endocardial-derived fibroblasts, mural or endothelial cells are rare. Unless the chick is profoundly different, this seems unlikely (and would require definitive proof). Just because they are not captured, that does not make them rare. The likely explanation is that the transition occurs rapidly and the cells don’t exist with a transitional (mixed) marker profile at the stages collected. As such, they would not be captured without lineage tracing or collection of additional, intermediate stages (they have only been captured as endocardium or derivatives, not mid-transition). Again, it is better to acknowledge technical limitations, than to draw potentially erroneous conclusions which risks misleading the field.

We agree with the interpretation of the reviewer, it is likely these cells are not captured because the transition states are short-lived. We have updated the manuscript accordingly.

5. The data on Thymosin b4 are also interpreted and discussed in a way that is misleading. Although they don't explicitly say so, as written, it is implied that there is little or no expression of Tmsb4x in cardiomyocytes. From their images and published literature, it is clear that there is expression in cardiomyocytes (in keeping with the known ubiquitous expression; only exception being erythrocytes). Qualitative ISH/ immunostaining always demonstrates relative expression; longer exposure would saturate vascular/epicardial signals but reveal abundant expression in cardiomyocytes. This is particularly apparent at the protein level (supplemental figure 9). As expected, protein expression in established cardiomyocytes would be less dynamic than mRNA in the actively developing epicardial and coronary lineages. In the discussion, the authors conclude that discrepancies in the literature reporting Tmsb4x mutant phenotypes may be explained by a failure to target the lineages that most abundantly express the gene. This is entirely unfounded. The studies cited targeted Tmsb4x globally and selectively in endothelium and myocardium (albeit with different methodologies) and still reached discrepant conclusions. Thus, global versus conditional targeting is not the explanation and the discussion requires rewording. Moreover, just because a gene is expressed at a lower level in one cell type compared with another does not necessarily mean a phenotype will be less severe if it is deleted. A further minor point, for clarification, the authors say that "one other beta-thymosin was expressed, Tmsb15" – were the others not expressed e.g. Tmsb9, Tmsb10?

Thank you for this comment and suggestions.

- We agree that our smFISH and immunostaining data shows TMSB4X expression (not enrichment) in cardiomyocytes. We have therefore explicitly stated that there is ubiquitous expression throughout all the cell types including cardiomyocytes to avoid any confusion.
- We have reworded the discussion and removed the claims suggesting "a failure to target the correct lineage cells".
- Only TMSB15B was expressed in our dataset (**Supplementary Fig. 7E**). Expression of TMSB9 and TMSB10 was not detected.

6. It would be helpful for those unfamiliar with the computational methods if the authors could briefly summarise the differences between the Monocle and PHATE for trajectory reconstruction.

Thank you for this suggestion. We have included this discussion as **Supplementary Note 1**.

7. Minor point: it is unclear what the authors mean by "significantly upregulated expression of midkine" in the epicardium. Up-regulation implies a change in expression (over time, with treatment etc.). Enriched would be more appropriate here to describe differential expression between cell types.

Thank you for the suggestion, we have made this correction in the new version of the manuscript.

Reviewer #3 (Remarks to the Author):

The authors addressed well the issues and concerns the reviewers had raised. Additional experiments and analyses conducted in the revised manuscript have improved the quality of that much. Now no major issues or concerns are pointed.

We thank the reviewer again for the great suggestions and comments, and for the appreciation of our work.

Only minor points: Please correct typos and misspelling, such as "PDGRFB" on Line 365, "PDGRB+" on Lines 368 and 369, etc.

Thank you for pointing this out, we have corrected these typos in the new version of the manuscript.